# Systematic identification of mutations and copy number alterations associated with cancer patient prognosis

**Joan C Smith[1], Jason M Sheltzer[2]***

[1]Google, Inc., New York, United States; [2]Cold Spring Harbor Laboratory, Cold Spring Harbor, United States

**Abstract** Successful treatment decisions in cancer depend on the accurate assessment of patient risk. To improve our understanding of the molecular alterations that underlie deadly malignancies, we analyzed the genomic profiles of 17,879 tumors from patients with known outcomes. We find that mutations in almost all cancer driver genes contain remarkably little information on patient prognosis. However, CNAs in these same driver genes harbor significant prognostic power. Focal CNAs are associated with worse outcomes than broad alterations, and CNAs in many driver genes remain prognostic when controlling for stage, grade, *TP53* status, and total aneuploidy. By performing a meta-analysis across independent patient cohorts, we identify robust prognostic biomarkers in specific cancer types, and we demonstrate that a subset of these alterations also confer specific therapeutic vulnerabilities. In total, our analysis establishes a comprehensive resource for cancer biomarker identification and underscores the importance of gene copy number profiling in assessing clinical risk.
DOI: https://doi.org/10.7554/eLife.39217.001

## Introduction

Cancers that arise from the same tissue can exhibit vast differences in clinical behavior. For instance, among individuals diagnosed with early-stage colorectal cancer, about 60% of patients will be cured by surgery alone, while the remaining 40% will experience a recurrence that is frequently fatal (*Mäkelä et al., 1995*). Various pathological and molecular biomarkers are typically analyzed in order to assess patient risk and aid clinical decision-making. In general, these biomarkers are divided into two classes: *predictive* and *prognostic* (*Nalejska et al., 2014*). Predictive biomarkers identify patients who are likely to respond to specific therapies, like the *EGFR* mutations that sensitize lung tumors to EGFR inhibition (*Paez et al., 2004*). In contrast, prognostic biomarkers provide information on cancer aggressiveness and the likelihood of patient death. Tumor de-differentiation and lymph-node infiltration serve as prototypical prognostic biomarkers due to their strong association with poor outcomes (*Connolly et al., 2003*). Yet, these pathology-based biomarkers can suffer from low levels of inter-observer concordance (*Allsbrook et al., 2001*; *Coons et al., 1997*; *Elmore et al., 2015*; *Gilks et al., 2013*), and even perfect pathological assessment yields incomplete information on a patient's most likely clinical course (*Bijker et al., 2013*; *Nofech-Mozes et al., 2005*; *Young, 2003*; *Zaniboni et al., 2004*). New methods to identify aggressive tumors could lead to improvements in the stratification of patient risk, better clinical management, and a decrease in dangerous and unnecessary over-treatment (*Esserman et al., 2013*).

Advances in high-throughput technologies have yielded unprecedented insight into the diverse array of genomic changes found within every cancer cell. Projects like The Cancer Genome Atlas (TCGA) and the International Cancer Genome Consortium (ICGC) have characterized methylation, mutation, copy number, and gene expression patterns across cancer types. As a result of these

***For correspondence:**
sheltzer@cshl.edu

**eLife digest** Cancers are not created equal: even when the disease affects the same organ, it can run different courses between individuals. For example, amongst people with early-stage bowel cancer who undergo surgery, 60% will go on to live cancer-free but the remaining patients will see the illness come back within a few years. These differences in outcome are still poorly understood, but they may find their roots in the genetic changes present in tumor cells.

Comparing the genomes of healthy and cancerous cells can help to understand which genetic modifications makes a cell go 'rogue' and start to multiply uncontrollably. Often, this happens because of a mutation, a change in the letters that compose our genetic code. However, looking at genetic differences between cancerous cells from different patients, or different tumors, can shed light on how certain genetic changes make the disease deadlier or more likely to reoccur.

Smith and Sheltzer looked into the genomes of 17,879 tumors from patients whose clinical information was also available. The analysis revealed that specific genetic alterations were more common in either deadly or treatable cancers. Most of these changes were not mutations that affected a few DNA letters; instead, they were copy number alterations, whereby large portions of the genetic code are being repeated or deleted. These results suggest that while mutations certainly drive the development of the disease, other changes such as copy number alterations can tell us which cancers will be deadlier. Through this approach, Smith and Sheltzer were also able to identify copy number alterations that were associated with patients responding well to certain drugs.

These findings now need to be confirmed on a different set of data. If they hold, new technologies may be developed so that the approach can be used cheaply and easily in the clinic. Ultimately, being able to examine copy number alterations in tumors may help physicians to tailor treatment for a particular cancer, or even a specific tumor.

DOI: https://doi.org/10.7554/eLife.39217.002

studies, many of the genomic differences between normal and transformed cells have been identified and characterized. However, we lack a similar understanding of the genomic differences between indolent tumors and aggressive malignancies. As the cost of DNA sequencing continues to drop, it has become increasingly feasible for hospitals to implement routine targeted and/or genome-wide analyses of patient tumors (*Gagan and Van Allen, 2015*; *Sholl et al., 2016*; *Zehir et al., 2017*). But, while several DNA-based, therapy-specific predictive biomarkers have been discovered, the prognostic information contained within tumor genomes is much less clear.

Previous genome-wide efforts to discover novel prognostic biomarkers have largely focused on the gene expression changes associated with patient mortality (*Anaya, 2016*; *Anaya et al., 2015*; *Gentles et al., 2015*; *Uhlen et al., 2017*). These studies have identified a set of transcripts that encode proteins involved in cell cycle progression that correlate with recurrence and death in several cancer types (*Cuzick et al., 2011*; *Dancik and Theodorescu, 2015*; *Gentles et al., 2015*; *Mosley and Keri, 2008*; *Venet et al., 2011*; *Wang et al., 2012*; *Wistuba et al., 2013*). Comparatively less is known about how changes at the DNA level affect patient survival. Outcome-associated analyses of genetic mutations have predominantly been conducted on a limited number of known oncogenes from single cancer types and have come to divergent conclusions. Reports in the literature commonly suggest that mutations in driver oncogenes are associated with poor outcomes, including, for instance, *KRAS* mutations in lung cancer (*Guan et al., 2013*; *Marabese et al., 2015*; *Sun et al., 2013*), *PIK3CA* mutations in breast cancer (*Li et al., 2006*; *Oshiro et al., 2015*), and *BRAF* mutations in colorectal cancer (*Richman et al., 2009*; *Roth et al., 2010*; *Tol et al., 2009*). Other studies of the same genes in the same cancer types have failed to observe any significant associations with outcome (*Bozhanov et al., 2010*; *Gonzalez-Angulo et al., 2009*; *Hutchins et al., 2011*; *Pang et al., 2014*; *Scoccianti et al., 2012*). In general, mutation-based biomarker studies may be confounded by small samples sizes, post-hoc hypothesis testing, imprecise clinical endpoints, and the so-called 'file drawer' problem, in which negative findings are less likely to be published (*Aronson, 2005*; *Ensor, 2014*; *Goossens et al., 2015*; *Rosenthal, 1979*; *Scargle, 1999*). The prognostic

information captured by sequencing driver oncogenes remains unknown, and a pan-cancer, exome-wide analysis of outcome-associated mutations has not been conducted.

Previous investigations into the prognostic importance of DNA copy number alterations (CNAs) have indicated that highly-aneuploid tumors tend to have worse outcomes than diploid tumors (*Friedlander et al., 1984*; *Kallioniemi et al., 1987*; *Kokal et al., 1986*; *Merkel and McGuire, 1990*; *Zimmerman et al., 1987*). However, these analyses have largely focused either on arm-length changes (*Davoli et al., 2017*; *Roy et al., 2016*) or on alterations that affect single oncogenes or tumor suppressors (*Deming et al., 2000*; *Shi et al., 2012*; *Srividya et al., 2011*). The functional importance of copy number changes in most genes at the single-gene level is unknown, and a pan-cancer, gene-by-gene analysis of prognostic copy number alterations has not been conducted. In order to gain a global understanding of the genomic features in a primary tumor that influence cancer prognosis, we collected and analyzed molecular profiles from a 'discovery' set of 9442 patients and a 'validation' set of 8618 patients with solid tumors. Our comprehensive, gene-centric analysis sheds light on the genomic changes that drive aggressive disease and will provide a useful resource for the development of strategies to improve clinical risk assessment. Additionally, we provide a web portal to facilitate community access to this rich biomarker dataset at http://survival.cshl.edu.

## A cross-platform, pan-cancer analysis of cancer survival data

To determine the differences between benign and fatal tumors, we first analyzed multiple classes of genomic data from 9442 patients with 16 types of cancer from the TCGA (outlined in *Figure 1—figure supplement 1A*; abbreviations are defined in *Figure 1—figure supplement 1B*). For every tumor type and every dataset, we generated Cox univariate proportional hazards models linking the presence or expression of a particular feature with clinical outcome (described in Supplemental Text 1). We report the Z score for each model, which encodes both the directionality and significance of a particular association. If the presence of a mutation or copy number amplification is significantly associated with patient death, then a Z score >1.96 corresponds to a P value < 0.05 (*Figure 1—figure supplement 2A–C*). In contrast, a Z score less than −1.96 indicates that the presence of a mutation is associated with survival or that a gene deletion is significantly associated with patient death.

We extracted mutation, copy number, gene expression, and clinical information from 16 TCGA cohorts (summarized in *Supplementary file 1* and discussed in additional detail in Supplemental Text 2). To assess the validity of our data analysis pipeline, as well as the accuracy of the reported patient annotations, we first examined the overall survival curves for the 16 tumor types that we profiled. As expected, we observed significant differences in clinical outcome according to a cancer's tissue-of-origin (*Figure 1—figure supplement 2D*). Prostate cancer had the least aggressive clinical course, with a median survival time that was not reached in this dataset (>4600 days), while pancreatic cancer conferred the worst prognosis (median survival time: 444 days). Overall, the 5 year survival frequencies of patients in the TCGA were highly similar to the national averages reported by NCI-SEER (R = 0.83, p < 0.0001), suggesting that the patients included in this analysis are broadly representative of the general population (*Figure 1—figure supplement 2E*). Next, we inferred patient sex on the basis of chromosome-specific gene expression patterns (*Gentles et al., 2015*; *van den Berge and Sijen, 2017*). Our analysis exhibited >99% concordance with patients' self-reported sex, further verifying the overall accuracy of the clinical annotations and our data processing pipeline (*Figure 1—figure supplement 2F*).

## Cancer mutations convey limited prognostic information

We first set out to discover whether coding mutations in cancer genomes were associated with patient outcome. We extracted non-silent mutations in each tumor, and then we identified all genes that were mutated in ≥2% of patients in each of the 16 cohorts (discussed in Supplemental Text 1). We next performed Cox proportional hazards analysis to compare the survival times for patients harboring mutant or wild-type copies of each gene. This analysis uncovered very few mutations that were significantly associated with patient outcome (*Figure 1* and *Supplementary file 2A-B*). We first focused on known oncogenes and tumor suppressors, and found that among the 30 most-frequently mutated cancer driver genes, only two (*EGFR* and *TP53*) were associated with prognosis in more than two tumor types (*Figure 1C*). *TP53* mutations were linked to outcome in five of 16 cancer types, though the differences in patient survival were generally small (*Figure 1—figure supplement 3A–B*).

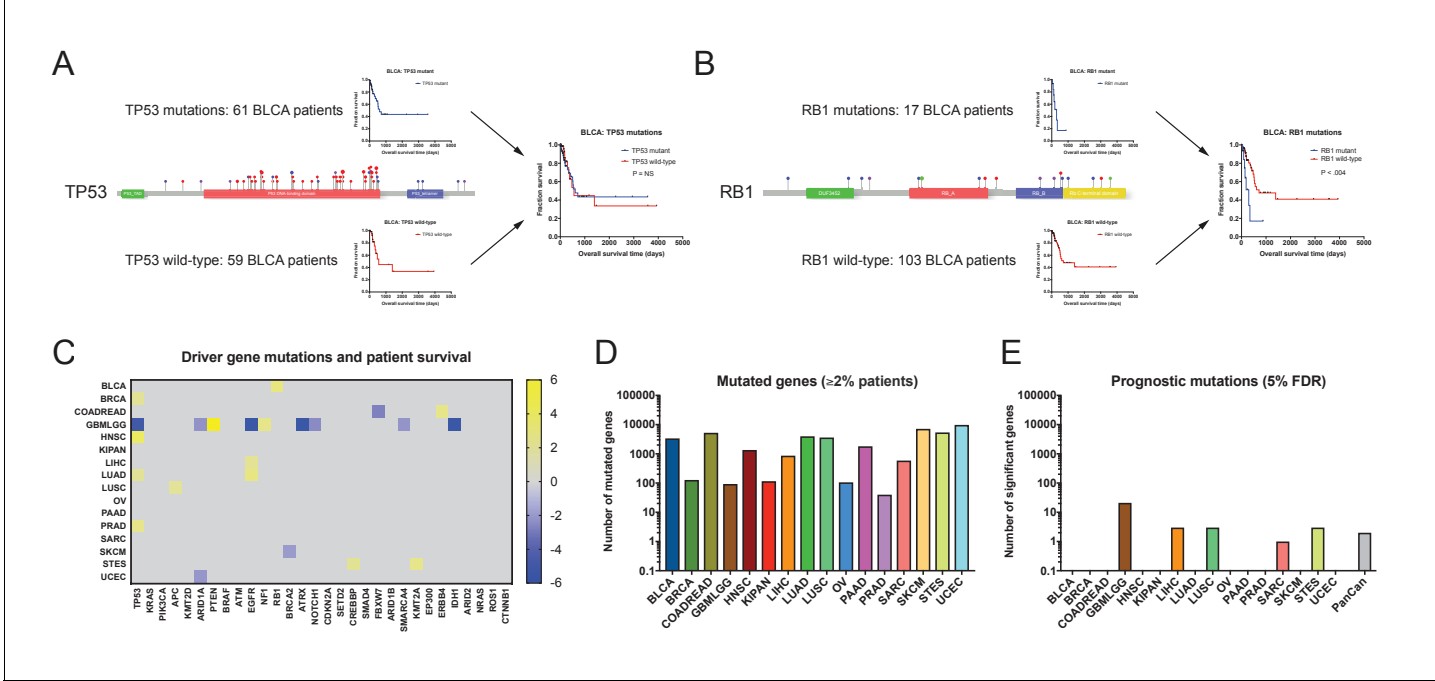

**Figure 1.** Single base-pair mutations convey limited prognostic information. (**A**) Schematic of TP53 mutations and patient survival in the BLCA patient cohort. Red dots indicate missense mutations, blue dots indicate frameshift mutations, and purple dots indicate nonsense mutations. (**B**) Schematic of RB1 mutations and patient survival in the BLCA patient cohort. Red dots indicate missense mutations, blue dots indicate frameshift mutations, purple dots indicate nonsense mutations, and green dots indicate splice-site mutations. Note that while 17 patients harbor RB1 mutations, 19 mutations are displayed on the lollipop plot, as two patients harbor two mutations in the RB1 gene. (**C**) A heatmap of significant survival associations among the 30 most frequently-mutated cancer driver genes in 16 tumor types from the TCGA are displayed. Z scores were calculated by regressing survival times between patients harboring wild-type and mutant copies of a gene if a gene was mutated in ≥2% of samples per tumor type. For visualization purposes, only significant Z scores are displayed. The complete list of Z scores is presented in *Supplementary file 2A*. (**D**) The number of genes mutated in ≥2% of samples per tumor type are displayed. (**E**) The number of genes significantly associated with patient outcome at a false-discovery threshold of 5% in each tumor type are displayed.

DOI: https://doi.org/10.7554/eLife.39217.003

The following figure supplements are available for figure 1:

**Figure supplement 1.** A schematic of the pan-cancer survival analysis pipeline and the datasets used.
DOI: https://doi.org/10.7554/eLife.39217.004

**Figure supplement 2.** Cox proportional hazards survival analysis and the accuracy of TCGA clinical annotations.
DOI: https://doi.org/10.7554/eLife.39217.005

**Figure supplement 3.** The mutation status of *TP53* is associated with outcome in multiple cancer types.
DOI: https://doi.org/10.7554/eLife.39217.006

**Figure supplement 4.** Hotspot mutations and mutations in multiple cancer driver genes are generally not associated with clinical prognosis.
DOI: https://doi.org/10.7554/eLife.39217.007

**Figure supplement 5.** Excluding patients with hypermutated tumors or those who were treated with targeted therapies fails to reveal mutations significantly associated with outcome.
DOI: https://doi.org/10.7554/eLife.39217.008

**Figure supplement 6.** Mutations with high variant allele frequencies are no more prognostic than mutations with low variant allele frequencies.
DOI: https://doi.org/10.7554/eLife.39217.009

**Figure supplement 7.** Prognostic mutations in glioma.
DOI: https://doi.org/10.7554/eLife.39217.010

In contrast, many other cancer driver genes were not associated with survival time in any tumor type. While mutations in *KRAS*, *PIK3CA*, *CDKN2A*, *BRAF*, *KMT2D*, *ATM*, *SMAD4*, and many other genes were frequently observed, they were never significantly linked with patient outcome (*Figure 1C*).

We next considered the possibility that mutations in specific codons could have prognostic significance not captured when all mutations in a gene are pooled together. To test this, we identified the

30 most-frequently mutated amino acid positions in the TCGA cohorts, and then asked whether patients harboring these alterations had different outcomes than those who did not. IDH1$^{c132}$ mutations were significantly associated with a favorable prognosis in glioma, but other recurrently-mutated codons (KRAS$^{c12}$, PIK3CA$^{c1047}$, TP53$^{c273}$, etc.) were largely uninformative (*Figure 1—figure supplement 4A–D*). Then, we identified 'hotspot' residues that were mutated in at least five different patients across all cohorts. Considering only these 'hotspot' mutations in each gene also failed to uncover robust survival associations (*Figure 1—figure supplement 4E*). Finally, we identified cancer type-specific recurrent mutations, but these alterations (FGFR3$^{c249}$ in BLCA, CTNNB1$^{c37}$ in UCEC, etc.) were similarly uninformative (*Figure 1—figure supplement 4F*).

Next, we sought to test whether the use of targeted therapies had blunted the deleterious effects of certain driver mutations (e.g., in *BRAF* or *EGFR*). However, due to the time-frame of sample collection, very few patients were treated with BRAF or EGFR inhibitors, and removing those patients who had received these therapies failed to significantly affect Z scores (*Figure 1—figure supplement 5A*). Hyper-mutation within a subset of cancers could increase mutational 'noise' and decrease our ability to identify prognostic signatures, but excluding patients with hyper-mutated tumors had minimal effect on the prognostic significance of driver gene mutations (*Figure 1—figure supplement 5B* and *Supplementary file 2C*). We then asked whether the presence of mutations in multiple cancer driver genes might cooperate to confer a worse clinical outcome. We found that, in general, patients harboring mutations in two cancer driver genes that were not prognostic alone had the same risk of death as patients with wild-type copies of one or both genes (*Figure 1—figure supplement 5C*). Lastly, we considered the possibility that the clonality of a mutation might affect its prognostic significance. We calculated the variant allele frequency (VAF) for each cancer mutation and tested whether mutations present at clonal levels in single tumors were more likely to be associated with outcome. We found that restricting our analysis to mutations with high VAFs failed to identify more prognostic genes, indicating that patient stratification is unlikely to be improved by assessing only clonal mutations (*Figure 1—figure supplement 6*).

These analyses suggested that, in general, cancer driver gene mutations lacked significant patient stratification power. This led us to investigate whether mutations in genes other than recurrently-mutated oncogenes and tumor suppressors could affect prognosis. We therefore expanded our analysis to include all genes mutated in ≥2% of patients with a particular tumor type. To account for greatly expanding the number of genes tested, we applied a Benjamini-Hochberg correction with a 5% false-discovery rate to the individual Z scores that we obtained. We uncovered several genes that were linked with prognosis in glioma, but found very few genes significantly associated with death or survival in the other 15 cancer types (*Figure 1D* and *Supplementary file 2A*). For instance, in breast cancer and lung adenocarcinoma, 128 and 3996 genes were mutated in ≥2% of patients, respectively, but none of these mutations were significantly correlated with patient outcome at a 5% FDR. In total, these results indicate that most mutations in cancer genomes lack significant prognostic power.

## Subtype-independent and subtype-dependent prognostic mutations in gliomas

In our above analysis, we noted that the five genes with the strongest survival associations were all observed in the GBMLGG (pan-glioma) cohort. As glioma appeared to be an exception to our overall finding that mutations are seldom prognostic, we investigated this cohort further. Among the top-scoring genes, we found that *PTEN* and *EGFR* mutations conferred dismal prognosis, while mutations in *IDH1*, *TP53*, and *ATRX* were associated with favorable prognosis (*Figure 1—figure supplement 7A*). Mutations in these genes have previously been linked to distinct glioma subtypes (*Ceccarelli et al., 2016*; *Kannan et al., 2012*; *Suzuki et al., 2015*), and we verified that mutations in *IDH1*, *TP53*, and *ATRX* were most frequently observed in low-grade gliomas, while mutations in *PTEN* and *EGFR* were most frequently observed in high-grade glioblastomas (*Figure 1—figure supplement 7B*). However, when we analyzed low-grade gliomas and glioblastomas separately, several of these alterations remained prognostic (*Supplementary file 2D*). For instance, while *IDH1* mutations were more common in low-grade gliomas, they were occasionally observed in high-grade tumors as well, and they were independently associated with prolonged survival in both cohorts (*Figure 1—figure supplement 7C*). In contrast, when *EGFR* mutations were observed in low-grade gliomas, they were associated poor outcomes, but *EGFR* mutations were non-prognostic in high-grade

glioblastomas (*Figure 1—figure supplement 7D*). Thus, in gliomas, mutations contain both subtype-dependent and subtype-independent prognostic information. However, outside of this cancer type and the tumor suppressor *TP53*, mutations in most cancer driver genes are non-prognostic.

## Driver gene CNAs are commonly associated with cancer patient mortality

As mutations were largely uninformative, we next set out to determine whether gene copy number conveyed prognostic information. We determined the copy number of each gene at its transcriptional start site and regressed this value against patient outcome in each tumor cohort. We then examined the clinical impact of CNAs affecting the same 30 cancer driver genes that we previously investigated. Surprisingly, we found that the copy number of these oncogenes and tumor suppressors was frequently linked with patient outcome (*Figure 2* and *Supplementary file 3A-B*). Amplification of *EGFR*, *PIK3CA*, and *BRAF*, and deletion of *CDKN2A*, *RB1* and *EP300* were strongly associated with shorter patient survival times in four or more cancer types each. Copy number was prognostic even for genes in which mutations were not linked with outcome: for instance, while mutations in *PIK3CA* were never informative, the copy number of *PIK3CA* was associated with outcome in breast, colorectal, glioma, lung-squamous, pancreas, and prostate cancers (*Figure 2B and D*). Overall, among the 30 most frequently-mutated cancer driver genes, we detected 108 significant associations between gene copy number and outcome, compared to 23 associations between mutation and outcome. For 28 out of 30 driver genes, DNA copy number was prognostic in more cancer types than mutational status was. We conclude that determining the copy number of oncogenes and tumor suppressors in a primary tumor can better stratify patient risk than assessing single base-pair mutations.

In our analysis thus far, we have treated mutations as a binary variable ('mutant' vs. 'not mutant'), while copy number alterations are treated as continuous values. Thus, the greater prognostic significance of tumor CNAs could reflect the fact that individual CNA measurements inherently harbor more information. To test this possibility, we trichotomized CNA values into 'deletions' ($<-0.3$), 'amplifications' ($>0.3$), and 'copy-neutral' ($\geq-0.3$ and $\leq0.3$). We then calculated Cox regressions at the same 30 loci using the discretized copy number values. This analysis resulted in 94 significant survival associations, more than four times as many significant features as when mutations were analyzed, and comparable to the number of significant features that resulted using continuous CNA values (*Figure 2—figure supplement 1*). This analysis suggests that the greater prognostic significance of CNAs is not simply a consequence of the continuous nature of copy number data.

We next investigated whether these oncogene and tumor suppressor CNAs were likely to drive patient mortality, or whether they were passenger genes that changed in copy number along with other, unknown drivers. To assess this question, we combined Z scores from different cancer types using Stouffer's method (*Stouffer, 1949*), and then plotted the pan-cancer meta-Z scores along every chromosome (*Figure 2C*). This analysis revealed multiple sharp peaks and valleys in the data that overlapped with known driver mutations. The most significant survival-associated copy number changes genome-wide were found on chromosome 9p in a valley that precisely included the tumor suppressor *CDKN2A*. Z score peaks were found at loci that include oncogenes *PIK3CA*, *EGFR*, *MYC*, *CCNE1*, and others. This overlap suggests that, in many instances, the copy number of these oncogenes and tumor suppressors directly influence the risk of cancer patient death.

## The prognostic significance of CNAs is independent of tumor sample purity and immune infiltration

Whole-chromosome aneuploidy has previously been linked to a decreased infiltration of immune cells (*Davoli et al., 2017*; *Taylor et al., 2018*). We therefore considered the possibility that CNAs are prognostic via an indirect mechanism; namely that they are found in tumors that lack robust immune infiltration, and this deficient immune response was itself driving patient mortality. However, multiple lines of evidence argue against this interpretation. First, we assessed the association between patient survival and three different measures of tumor sample purity: pathologist-assessed tumor cell fraction, sample purity as judged by ABSOLUTE (*Carter et al., 2012*; *Taylor et al., 2018*), and leukocyte infiltration, as judged by methylation analysis (*Taylor et al., 2018*). We found that sample purity was inconsistently-associated with patient outcome (*Figure 2—figure supplement 2*).

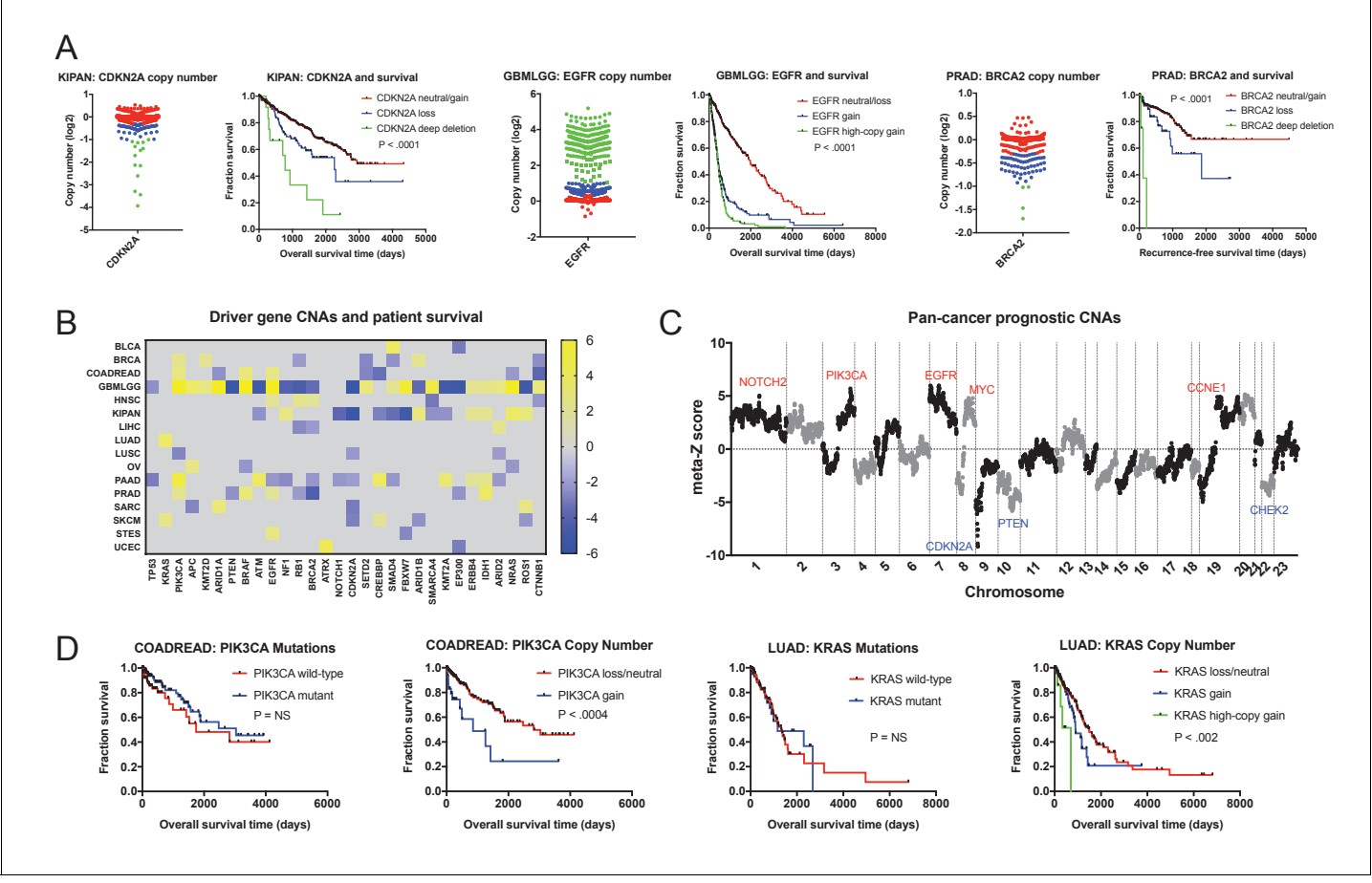

**Figure 2.** Oncogene and tumor suppressor CNAs drive cancer patient mortality. (**A**) Examples of driver gene CNAs associated with patient outcome. The copy number of *CDKN2A*, *EGFR*, and *BRCA2* in the indicated patient cohorts are displayed, as well as Kaplan-Meier curves of patient survival according to gene copy number. Amplifications and deletions correspond to CNAs > |0.3|, while deep-deletions and high-copy gains correspond to CNAs > |1|. (**B**) A heatmap of significant survival associations among the 30 most frequently-mutated cancer driver genes in 16 tumor types from the TCGA are displayed. Z scores were calculated by regressing gene copy number against patient outcome within each tumor type. The complete list of Z scores is presented in *Supplementary file 3A*. (**C**) Z scores from 16 cancer types from the TCGA were combined using Stouffer's method, and then the resulting meta-Z scores were plotted against the chromosomal location. Genes were binned by average Z score into groups of 5 for visualization. Gene names indicate candidate driver genes found within survival-associated peaks and valleys. (**D**) Kaplan-Meier curves are plotted for two oncogenes, *PIK3CA* (left) and *KRAS* (right), comparing the prognostic relevance of mutations in those genes versus copy number alterations in these genes. Amplifications correspond to CNAs > 0.3, while high-copy gains correspond to CNAs > 1.

DOI: https://doi.org/10.7554/eLife.39217.011

The following figure supplements are available for figure 2:

**Figure supplement 1.** Discretized copy number values still hold significant prognostic power.
DOI: https://doi.org/10.7554/eLife.39217.012

**Figure supplement 2.** The prognostic value of cancer CNAs is independent of tumor sample purity.
DOI: https://doi.org/10.7554/eLife.39217.013

**Figure supplement 3.** CNAs remain prognostic after correcting for tumor stage and grade.
DOI: https://doi.org/10.7554/eLife.39217.014

**Figure supplement 4.** CNAs remain prognostic after correcting for tumor subtype.
DOI: https://doi.org/10.7554/eLife.39217.015

For instance, higher tumor purity determined by either pathological analysis or ABSOLUTE was associated with worse outcome in only one of 16 cohorts, each (*Figure 2—figure supplement 2A–B*). The lack of a strong correlation between infiltrating cell populations and clinical prognosis suggests that analyte purity is insufficient to explain the relationship between CNAs and patient survival. Secondly, we generated multivariate Cox models that included gene copy number and these three

measurements of tumor purity, and we found that driver gene CNAs remain broadly prognostic in these bivariate models (*Figure 2—figure supplement 2C* and *Supplementary file 3C-E*). For instance, we discovered that amplification of Cyclin E1 is associated with poor prognosis in ovarian cancer, and this remained true even when our analysis was restricted to high-purity tumor samples and samples that lacked significant leukocyte presence (*Figure 2—figure supplement 2D*). Thus, while the interrelationship between aneuploidy and immunological tolerance likely plays an important role in tumor development, this analysis suggests that it is not the primary driver of CNA-associated patient mortality.

## Copy number analysis improves on patient stratification conferred by common clinical parameters

Pathological assessment of tumor stage and grade are important sources of prognostic information, though blinded assessments reveal significant inter-observer discordance (*Allsbrook et al., 2001*; *Coons et al., 1997*; *Elmore et al., 2015*; *Gilks et al., 2013*). We therefore tested whether the CNA biomarkers that we uncovered could affect the stratification conferred by these parameters. We found that Z scores generated from either univariate models or multivariate models that included stage or grade were highly correlated (R = 0.91 and R = 0.96 respectively; *Figure 2—figure supplement 3A* and *Supplementary file 4*). Overall, 71% of prognostic CNAs in individual cancer types remained prognostic in these multivariate models (*Figure 2—figure supplement 3B*). Thus, including gene-level copy-number assessment can significantly improve the stratification of patient risk beyond standard clinical parameters (*Figure 2—figure supplement 3C–D*). Certain mutations were similarly able to yield prognostic information in a stage- and grade-independent manner. However, due to the lower overall significance of most mutations that we identified, the improvements in patient stratification were generally more modest (*Figure 2—figure supplement 3E* and *Supplementary file 4E*).

Gene-level copy number values also remained prognostic when separating TCGA cohorts by cancer subtype (*Figure 2—figure supplement 4* and *Supplementary file 4F-G*). For instance, CNA Z scores values were highly correlated between the bulk GBMLGG cohort and the individual GBM and LGG subtypes (R = 0.68 and R = 0.86, respectively; *Figure 2—figure supplement 4A*). While analyzing the GBM cohort separately abolished the prognostic significance of *EGFR* mutations (*Figure 2—figure supplement 4D*), *EGFR* amplifications remained associated with outcome in both the LGG and GBM cohorts (*Figure 2—figure supplement 4B–C*). Amplifications in *MYC* and *PIK3CA* were similarly prognostic in multiple tumor subtypes (*Figure 2—figure supplement 4D–E* and *Supplementary file 4G*). At other loci, low patient numbers from certain subtypes may obscure the detection of specific biomarkers. For instance, within the KIPAN cohort, 67% of tumors are clear cell carcinomas, 23% of tumors are papillary cell carcinomas, and 10% of tumors are chromophobe carcinomas. *CDKN2A* deletion is a strong indicator of poor prognosis in the pan-kidney cohort, in clear cell carcinomas, and in papillary cell carcinomas, but did not reach statistical significance when kidney chromophobe carcinomas were analyzed independently (*Figure 2—figure supplement 4F–G*). In total, these results underscore the ability of driver gene CNAs to improve patient stratification when controlling for tumor identity, though larger cohort numbers may be needed to identify the strongest biomarkers in rare cancer subtypes.

## Driver gene CNAs contain prognostic information not captured by *TP53* mutation status or total aneuploidy

Highly-aneuploid tumors tend to harbor mutations in *TP53*, and both *TP53* mutations and arm-length aneuploidy have previously been associated with poor clinical outcomes (*Davoli et al., 2017*; *Petitjean et al., 2007*). Using an 'aneuploidy score' for each tumor based on the total number of arm-length alterations (*Taylor et al., 2018*), we verified that *TP53*-mutant tumors exhibit more aneuploidy than *TP53*-wild-type tumors (*Figure 3—figure supplement 1A*), and that total aneuploidy is a poor prognosis factor in several cancer types (*Figure 3—figure supplement 1C*). To investigate the relationship between gene-level prognostic CNAs, *TP53* status, and arm-length aneuploidy, we selected a set of 40 prognostic amplifications and deletions for additional analysis (*Figure 3—figure supplement 2A*). In multivariate models that included *TP53* mutation status, 33 of 40 (83%) gene-level CNAs remained prognostic, demonstrating that these CNAs are not linked with death due to

an indirect association with *TP53* status (*Figure 3—figure supplement 2A–B*). Similarly, in multivariate models that included total tumor aneuploidy, 80% of these CNAs were still associated with outcome (*Figure 3—figure supplement 2C–D*). Finally, as a proxy for the total structural alteration burden, we summed the number of breakpoints (as indicated by discrete copy number values along a chromosome) in each tumor (*Figure 3—figure supplement 1B*). This metric was associated with outcome in multiple tumor types (*Figure 3—figure supplement 1C*), but 75% of driver gene CNAs remained prognostic in multivariate models that included this score (*Figure 3—figure supplement 2E–F*). These results indicate that assessing gene-level tumor CNAs can yield more prognostic information than simply screening for *TP53* mutations or measuring bulk levels of tumor aneuploidy (*Supplementary file 5*).

## Focal CNAs typically portend worse prognosis than broad CNAs

We next set out to determine whether focal copy number alterations and broad copy number alterations could have distinct effects on patient outcome. To investigate this possibility, we compared the prognostic power of focal CNAs (defined as an alteration ≤3 Mb in length; *Krijgsman et al., 2014*) and broad CNAs (defined as all alterations >3 Mb in length). Among loci at which both broad and focal alterations were observed, we frequently found that broad CNAs were associated with moderately worse outcomes, while focal CNAs were associated with sharp declines in survival (*Figure 3A–B*). At some loci, broad CNAs had outcomes that were indistinguishable from copy-neutral tumors, while only focal CNAs were associated with death (*Figure 3C*). We rarely detected instances in which broad CNAs indicated a worse prognosis than a focal alteration (*Figure 3A*). We interpret these results as a reflection of aneuploidy-induced fitness penalties (*Sheltzer et al., 2017*; *Sheltzer and Amon, 2011*): large copy number alterations change the dosage of multiple genes at once and can impair tumor growth, while targeted alterations that specifically affect driver gene copy number maximize malignant potential.

## Focal CNAs affect patient outcome by changing the expression levels of wild-type genes

Gene copy number alterations typically result in a proportional change in the expression of the affected loci (*Pollack et al., 2002*; *Stingele et al., 2012*; *Williams et al., 2008*), though instances of dosage compensation have been reported (*Gonçalves et al., 2017*). To test the effects of prognostic CNAs on gene expression, we compared transcript levels and gene copy number changes at 40 prognostic loci and found a significant correlation between the two at 98% of the analyzed genes (*Figure 3—figure supplement 3*). Next, we sought to uncover whether these copy number alterations were deadly because they increased or decreased the expression of mutant gene products. That is, we could observe that the amplification of a driver gene is prognostic only in tumors in which that driver gene is also mutated. Interestingly, this is not the case: at 95% of our test loci, gene copy number remained prognostic in multivariate models that also included gene mutation status (*Figure 3D*). For instance, in colorectal cancer, amplification of *EGFR* was associated with death even in tumors that lacked *EGFR* mutations (*Figure 3E*). In total, these results indicate that even at recurrently-mutated loci, changes in the expression of the wild-type gene can have a profound effect on cancer cell behavior. Together with our observation that focal changes tend to confer a worse prognosis than broad changes, these results support the recently-proposed 'cancer gene island' model of tumor genome evolution (discussed in more detail below; *Solimini et al., 2012*).

## Independent patient cohorts verify the prognostic significance of driver gene CNAs

To determine the generality of our findings, we collected independent patient cohorts harboring mutation or copy number data linked to survival outcome (*Supplementary file 1*). We then performed univariate Cox proportional hazards analysis on these 'validation' cohorts and compared the results to the Z scores obtained from our 'discovery' set of TCGA data. First, we identified prognostic mutations within a set of 16 patient cohorts from the International Cancer Genome Consortium (ICGC), comprising 3054 patients analyzed by whole-genome or whole-exome sequencing. Overall, the mutation frequencies and the Z scores of recurrent mutations were highly similar between the ICGC and TCGA cohorts (R = 0.67, p < 0.0001, and R = 0.56, p < 0.0001, respectively; *Figure 4A–*

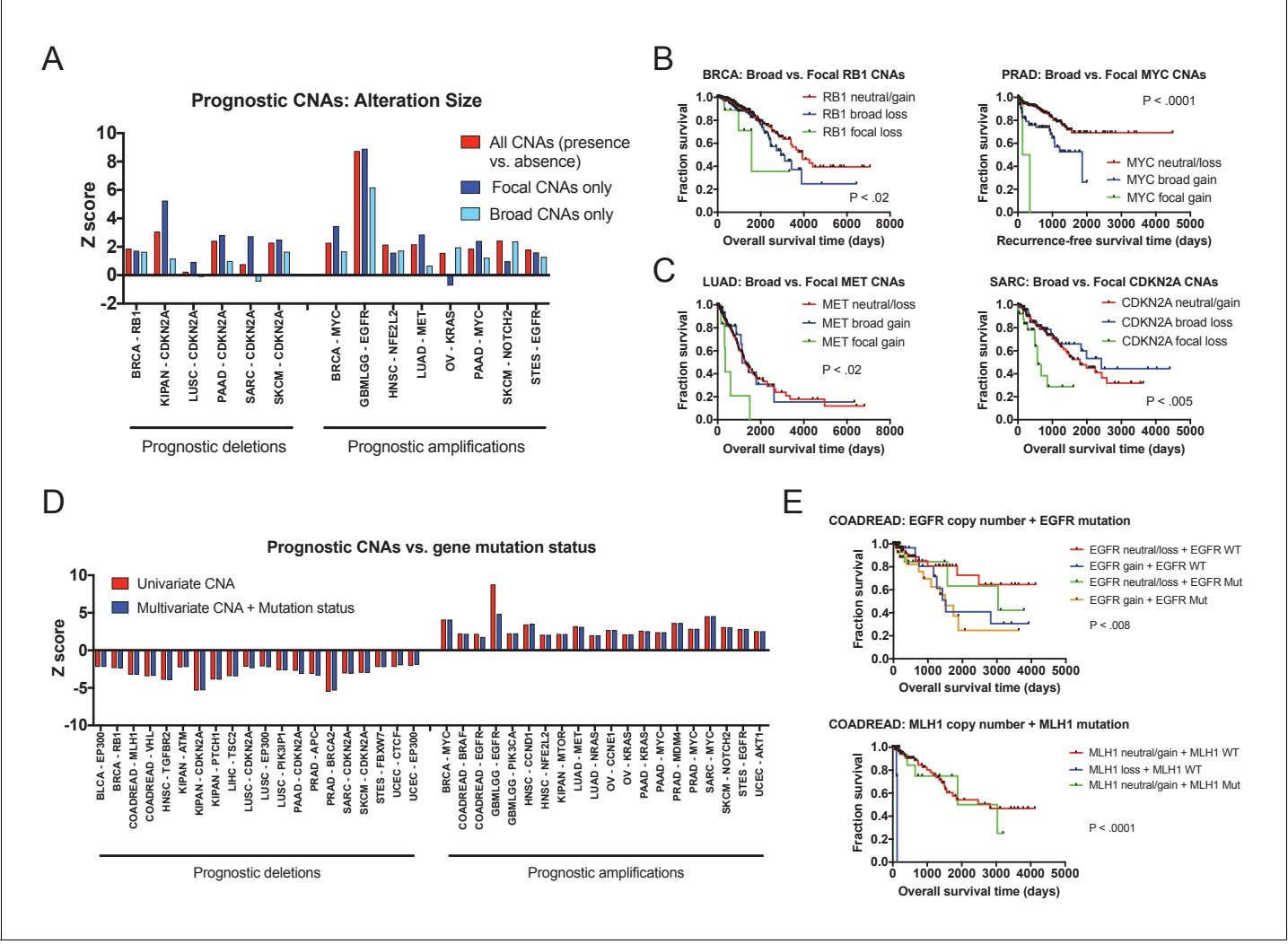

**Figure 3.** Effects of amplicon size and gene mutation status on prognostic CNAs. (**A**) 20 prognostic amplifications and 20 prognostic deletions were selected for further analysis (see also *Figure 3—figure supplement 2*). Of those 40, 14 had at least five patients who had focal CNAs (≤3 Mb) and at least five patients who had broad CNAs (>3 Mb). Univariate Cox proportional hazards models were constructed comparing the presence or absence of any CNA at the indicated locus, or comparing the presence or absence of a CNA of a particular size. (**B and C**) Kaplan-Meier curves are plotted at four prognostic loci comparing tumors with focal CNAs (≤3 Mb), tumors with broad CNAs (>3 Mb), and tumors that lack CNAs at that locus. Amplifications and deletions correspond to CNAs > |0.3|. (**D**) Multivariate Cox proportional hazards models were constructed including both the copy number of the indicated gene as well as the mutational status of that gene. Z scores for either the univariate models (CNAs alone) or the multivariate models (CNAs + mutation status) are displayed. (**E**) Kaplan-Meier curves comparing gene mutation status and gene copy number for *EGFR* and *MLH1* alterations in colorectal cancer. *EGFR* amplification and *MLH1* deletion are associated with poor prognosis, regardless of whether the tumor harbors an *EGFR* or *MLH1* mutation. In the bottom graph, note that no tumors harbored both *MLH1* deletions and mutations.
DOI: https://doi.org/10.7554/eLife.39217.016

The following figure supplements are available for figure 3:

**Figure supplement 1.** Gene-level CNAs, *TP53* status, total tumor aneuploidy, and total alteration burden.
DOI: https://doi.org/10.7554/eLife.39217.017
**Figure supplement 2.** Multivariate analysis of prognostic CNAs.
DOI: https://doi.org/10.7554/eLife.39217.018
**Figure supplement 3.** Prognostic CNAs alter the expression of the gene that they encompass.
DOI: https://doi.org/10.7554/eLife.39217.019

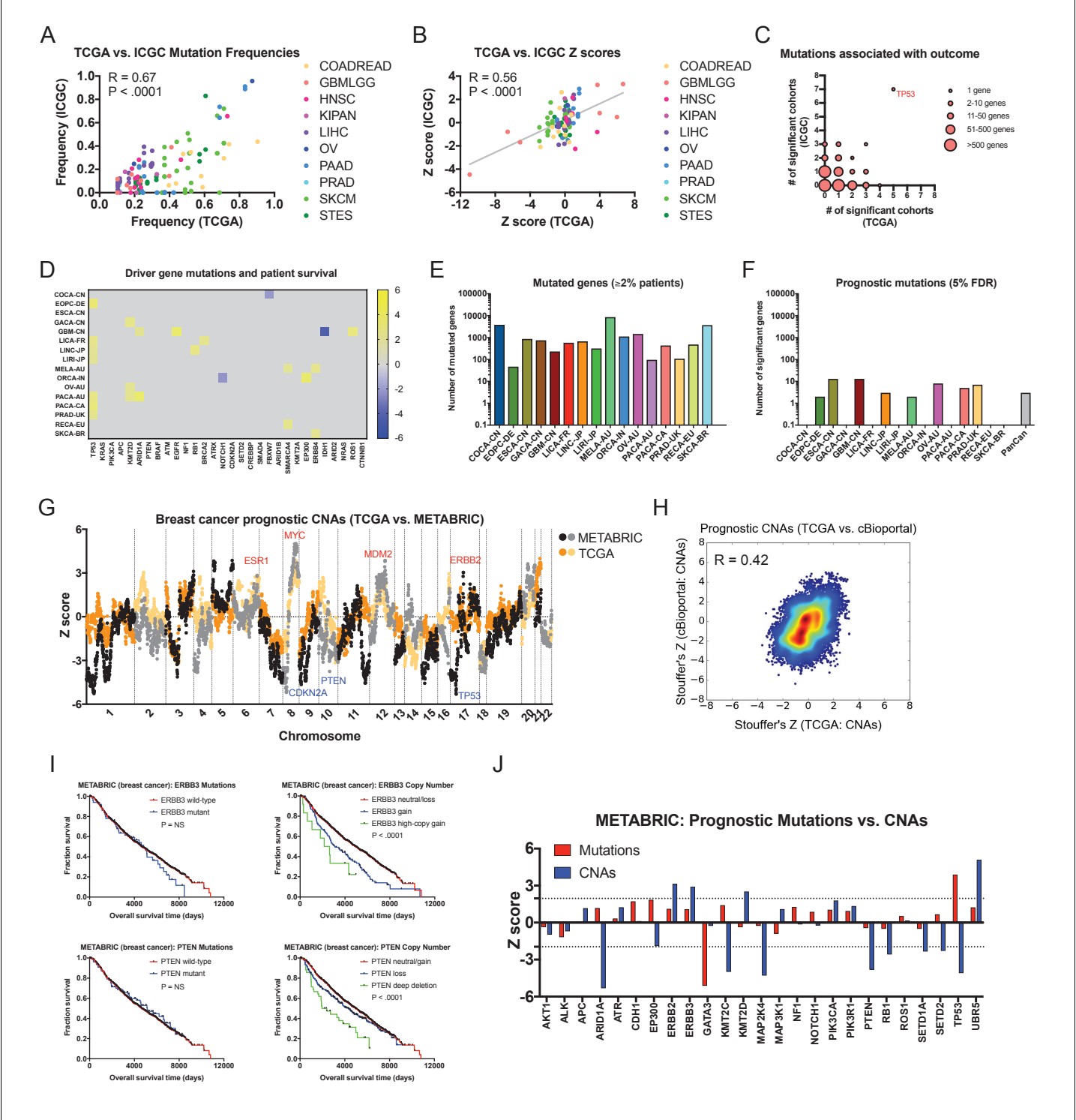

**Figure 4.** Driver gene copy number, but not driver gene mutations, are associated with survival in independent patient cohorts. (**A**) Genes mutated in ≥10% of patients in each tumor type in the TCGA were identified, and then compared to the mutation frequency of these genes in the corresponding ICGC cohort or cohorts. The complete list of Z scores is presented in *Supplementary file 6A*. (**B**) Z scores of the 10 most frequently-mutated genes per cancer type in the ICGC were identified and then plotted against the Z scores of the same gene from the corresponding TCGA cohort or cohorts. (**C**) Significant Z scores (>1.96 or<−1.96) were counted per gene, and then the number of significant cohorts from the TCGA and the ICGC are plotted. While the vast majority of frequently-mutated genes are significant in zero or one cancer type, *TP53* mutation status is associated with prognosis in 12 of 32 total patient cohorts. (**D**) A heatmap of significant survival associations among the 30 most frequently-mutated cancer driver genes in 16 patient cohorts from the ICGC are displayed. Z scores were calculated by regressing survival times between patients harboring wild-type

*Figure 4 continued on next page*

*Figure 4 continued*

and mutant copies of a gene if a gene was mutated in ≥2% of samples per tumor type. For visualization purposes, only significant Z scores are displayed. The complete list of Z scores is presented in *Supplementary file 6A*. (E) The number of genes mutated in ≥2% of samples per tumor type are displayed. (F) The number of genes significantly associated with patient outcome at a false-discovery threshold of 5% in each tumor type are displayed. (G) Z scores for the copy number of each gene from the TCGA BRCA cohort and the cBioportal METABRIC cohort are plotted against one another. The complete list of Z scores is presented in *Supplementary file 6C*. (H) Meta-Z scores from datasets curated by cBioportal are plotted against meta-Z scores from the corresponding four cancer types from TCGA (BLCA, BRCA, LIHC, and LUAD). The complete list of Z scores is presented in *Supplementary file 6C*. (I) Kaplan-Meier curves comparing mutations and CNAs in *ERBB3* and *PTEN* in the cBioportal METABRIC cohort. (J) A bar graph of Z scores for mutations and CNAs in 25 driver genes in the cBioportal METABRIC cohort. While mutations in only two genes are associated with prognosis, CNAs in 12 of these same genes are associated with prognosis.

DOI: https://doi.org/10.7554/eLife.39217.020

*B*). Consistent with our TCGA analysis, mutations in *TP53* were associated with outcome in more patient cohorts than any other gene (*Figure 4C* and *Figure 1—figure supplement 3C–D*). Other mutations, including in known cancer driver genes, were rarely associated with outcome in individual cancer types and harbored minimal pan-cancer significance (*Figure 4D–F* and *Supplementary file 6*). Mutations in *KRAS*, *PIK3CA*, *BRAF*, *APC*, *PTEN*, *CDKN2A*, and many others were frequently observed but were never correlated with outcome (*Figure 4D*). We next analyzed 2431 additional patients with CNA data curated by cBioportal, and found numerous amplifications and deletions associated with patient mortality (*Supplementary file 6C*). In breast cancer, we found prognostic amplifications that were centered around oncogenes, including *ERBB2*, *MYC*, and *MDM2*, while prognostic deletions encompassed tumor suppressors *CDKN2A*, *PTEN*, and *TP53* (*Figure 4G*). Overall, we observed a highly significant correlation between the meta-Z scores obtained from the TCGA and cBioportal datasets (R = 0.42; *Figure 4H*). Finally, in patient cohorts subjected to both mutation and copy number analysis, we verified that CNAs in driver genes commonly harbored greater prognostic significance than mutations in those same genes (*Figure 4I*). For instance, in breast cancer, among 25 frequently-mutated genes, mutations in only two genes (*TP53* and *GATA3*) displayed prognostic significance, while CNAs in 12 of those same genes were associated with patient outcome (*Figure 4J*). In total, these analyses suggest that the survival patterns discovered in the TCGA dataset are conserved across independent cohorts of cancer patients. In particular, while mutations in most cancer driver genes are non-prognostic, copy number alterations in these same genes are tightly linked with patient outcome.

## Cross-cohort identification of high-confidence prognostic biomarkers

In order to discover the biomarkers with the greatest potential clinical relevance, we next identified the individual mutations and CNAs that were consistently associated with outcome across independent patient cohorts. To increase our ability to detect these genetic alterations, we performed survival analysis on an additional set of 2701 primary tumors subjected to targeted sequencing and copy number analysis (MSKCC_2017; *Supplementary file 7*) (*Zehir et al., 2017*), on 2431 patients from cBioportal cohorts whose tumors had been sequenced (*Supplementary file 6D*), and on 628 patients from ICGC cohorts subjected to copy number analysis (*Supplementary file 6B*). Our combined patient dataset therefore included two to six independent cohorts from each of 13 common cancer types, comprising 16,580 total patients. These cohorts were collected at different locations, in different patient populations, using different study designs, and the samples were analyzed using different genomic technologies. We reasoned that alterations that were consistently associated with outcome despite these significant differences would represent highly-penetrant biomarkers of patient prognosis. To identify such alterations, we screened for biomarkers that were associated with outcome ($|Z| > 1.96$) in ≥2 independent cohorts, and that were highly significant ($|meta-Z| > 3.3$) across all available cohorts. This approach revealed multiple high-confidence genetic biomarkers of patient outcome that, to our knowledge, were novel, including *MDM4* amplifications in prostate cancer, *NOTCH2* amplifications in melanoma, and 2q32 deletions in ovarian cancer (*Supplementary file 8*). These robust biomarkers allowed a striking stratification of patient risk, and top-scoring CNAs remained prognostic in multivariate models that included commonly-measured prognostic criteria (Gleason score in prostate cancer, Hepatitis serology in liver cancer, etc.; *Figure 5—figure supplement 1*). Consistent with our single-cohort analyses, cross-cohort prognostic CNAs were significantly

more common than prognostic mutations, and *TP53* was the only gene whose mutation status was associated with outcome in more than one cancer type (*Figure 5—figure supplement 2A*).

## Certain prognostic biomarkers are also associated with unique therapeutic vulnerabilities

We hypothesized that some genetic alterations that were sufficient to affect overall patient survival could impact other facets of cancer behavior as well, including, potentially, drug sensitivity. That is, biomarkers harboring significant *prognostic* information could potentially contain *predictive* information as well. We therefore sought to discover whether genetic alterations that drove aggressive disease could also sensitize patient tumors to specific therapeutic regimens. By analyzing a cohort of 1000 patient-derived xenografts (PDXs), we identified several instances in which high-confidence biomarkers were associated with vulnerability to particular anti-cancer agents (*Gao et al., 2015a*). For instance, we identified Chr9 deletions that encompassed CDKN2A as a robust biomarker for poor prognosis in breast cancer (*Supplementary file 8*). We found that PDXs harboring CDKN2A deletions were profoundly sensitive to combination therapy with a CDK4/6 inhibitor and an mTOR inhibitor (*Figure 5—figure supplement 2B*), consistent with the fact that a protein encoded by *CDKN2A*, p16, functions as a natural inhibitor of CDK4/6 (*Serrano et al., 1993*), p. 4). In contrast, other biomarkers associated with poor prognosis in breast cancer failed to predict sensitivity to this treatment combination, but instead correlated with sensitivity to other agents (*Supplementary file 8*). Due to the limited number of drugs tested in PDXs, we expanded our target search to include a recently-described pharmacogenomic profile of cancer cell lines and discovered several additional biomarker vulnerabilities (*Figure 5A–B*). For instance, we identified mutations in *STAG2* as a high-confidence biomarker of poor prognosis in glioma, and we found that *STAG2*-mutant gliomas were exquisitely sensitive to treatment with the PARP inhibitor olaparib (*Figure 5A*). In total, we identified highly-significant therapeutic vulnerabilities for 49% of the prognostic biomarkers uncovered by our integrated analysis, providing potential strategies to treat a subset of patients who have the most aggressive cancers.

## Discussion

Modern medicine has vastly prolonged the survival of individuals diagnosed with cancer (*Johnson et al., 2017*). However, increasing evidence suggests that large subsets of patients receive sub-optimal care, and are over-treated or under-treated relative to their level of risk (*Bhatt and Klotz, 2016*; *Esserman et al., 2013*; *Swaminathan and Swaminathan, 2015*). To date, many of the genetic alterations that differentiate fatal and benign tumors have remained obscure. Our analysis of prognostic biomarkers from 17,879 patients sheds light on these genetic differences, identifies a subset of patients who may benefit the most from aggressive intervention, and suggests therapeutic strategies for tumors harboring certain alterations associated with poor prognosis. A web portal to facilitate access to these results is available at http://survival.cshl.edu/.

As cancers arise due to the accumulation of mutations in growth-promoting oncogenes and growth-inhibitory tumor suppressors, the presence and diversity of these mutations may be expected to dictate a tumor's clinical course. However, our data suggest that in many cases, they do not. Substantial disagreements exist in the literature on the value of mutation-based prognostic biomarkers, as the same driver oncogenes have been independently reported to be either adverse or non-significant prognostic features (*Guan et al., 2013*; *Marabese et al., 2015*; *Scoccianti et al., 2012*; *Sun et al., 2013*). In this manuscript, we performed an unbiased genome-wide analysis of public datasets with pre-established sample sizes. This approach may therefore bypass certain problems, including post-hoc hypothesis testing, patient-selection bias, and the 'file-drawer problem', that can confound targeted biomarker studies (*Aronson, 2005*; *Ensor, 2014*; *Goossens et al., 2015*; *Rosenthal, 1979*; *Scargle, 1999*). We consider it possible that, with larger sample sizes or more-specific tumor subtypes, additional prognostic mutations could be identified. Importantly, in most patient cohorts that we collected, tumors were analyzed on multiple genomic platforms, and CNAs were commonly prognostic in the same cohorts in which gene mutations were not. These results underscore our ability to successfully detect biomarkers in cohorts of these sizes, and suggest that, in a head-to-head comparison, copy number alterations provide more useful prognostic information than single-gene mutations.

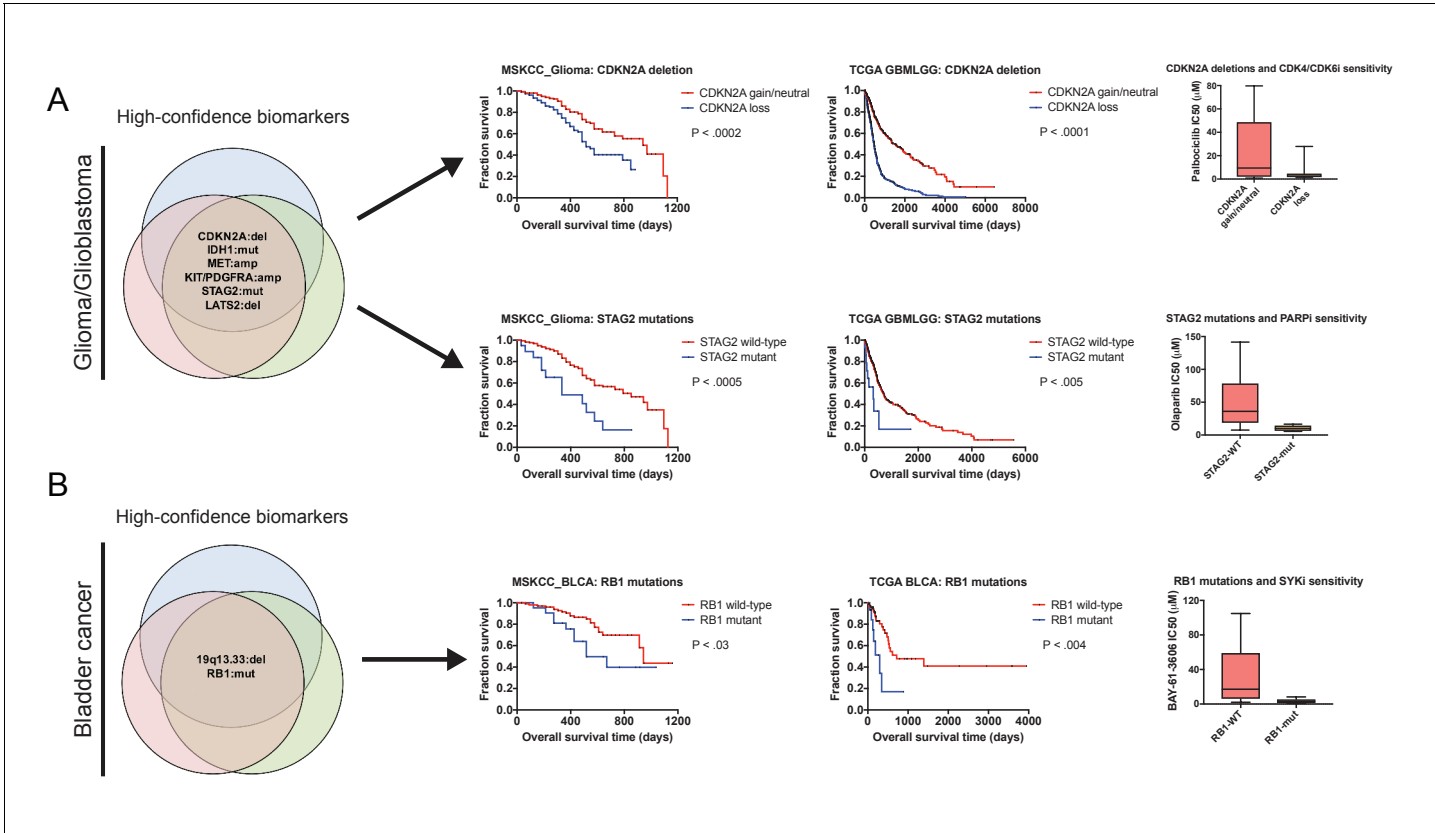

**Figure 5.** Robust prognostic biomarkers associated with drug sensitivity in cancer cell lines. (**A**) Mutations and CNAs associated with patient outcome in multiple cohorts of glioma/glioblastoma are displayed. Mutations in *STAG2* are associated with sensitivity to the PARP inhibitor olaparib, while CDKN2A deletions are associated with sensitivity to the CDK4/6 inhibitor palbociclib in glioma cell lines (*Iorio et al., 2016*). (**B**) Mutations and CNAs associated with patient outcome in multiple cohorts of bladder cancer are displayed. Mutations in *RB1* are associated with sensitivity to the SYK inhibitor BAY-61–3606 in bladder cancer cell lines (*Iorio et al., 2016*). The complete list of high-confidence biomarkers and potential vulnerabilities are listed in *Supplementary file 8*.

DOI: https://doi.org/10.7554/eLife.39217.021

The following figure supplements are available for figure 5:

**Figure supplement 1.** Multivariate analysis of high-confidence biomarkers with standard clinical criteria.

DOI: https://doi.org/10.7554/eLife.39217.022

**Figure supplement 2.** Robust prognostic biomarkers associated with drug sensitivity in cancer cell lines.

DOI: https://doi.org/10.7554/eLife.39217.023

While we identified very few mutations associated with patient outcome, several lines of evidence underscore the potential benefits of continued clinical sequencing efforts. First, our analysis revealed a subset of mutations with tissue-specific prognostic power, including *TP53* mutations in breast cancer, *RB1* mutations in bladder cancer, and *FBXW7* mutations in colorectal cancer. Secondly, most patients in the TCGA cohorts were treated with standard cytotoxic drugs. As targeted and immunotherapies are increasingly adopted in the clinic, oncogenic mutations that were non-prognostic in the datasets analyzed here may be able to predict sensitivity to specific therapeutic agents (*Gagan and Van Allen, 2015*). Thirdly, tumors themselves are composed of sub-clonal populations that harbor distinct sets of mutations, and recent evidence suggests that cancer heterogeneity can influence clinical course (*Jamal-Hanjani et al., 2017*). Thus, interrogating the mutational spectrum at the sub-clonal level may identify prognostic mutations not distinguished in bulk analyses.

Though large-scale changes in tumor ploidy have previously been recognized as an indicator of poor outcome (*Friedlander et al., 1984*; *Kallioniemi et al., 1987*; *Kokal et al., 1986*; *Merkel and McGuire, 1990*; *Zimmerman et al., 1987*), the contributions of copy number alterations in most single genes have remained unexplored. Despite the limited stratification value of mutations in cancer

driver genes, we found that copy number alterations of many of these same genes are broadly prognostic. Focal CNAs tended to confer a worse prognosis than broad CNAs, consistent with a model in which large-scale gene dosage imbalances trigger proteotoxic stress and impose a fitness penalty on cancer cells (*Santaguida and Amon, 2015*; *Sheltzer and Amon, 2011*). Moreover, while prognostic CNAs commonly caused proportional changes in target gene expression, most CNAs remained prognostic whether or not they affected the expression of a mutated gene. These results support a 'cancer gene island' or 'cumulative aneuploidy' model of tumorigenesis, in which cancers accumulate a series of limited copy number changes affecting haplo-sensitive and triplo-sensitive regions (*Davoli et al., 2013*; *Solimini et al., 2012*). Identifying the functional consequences of these prognostic CNAs on tumor physiology is a key future goal.

Patients whose tumors harbor genetic alterations that drive mortality are in urgent need of improved treatment options. We discovered many instances in which high-confidence biomarkers of aggressive disease also sensitized tumors to specific anti-cancer therapies. By taking advantage of these vulnerabilities, a precision-medicine approach could be applied to both stratify patient risk and identify drug combinations most likely to provide a clinical benefit. Several predicted sensitivities from our work have clinical or mechanistic support, including the use of CDK4/6 inhibitors to treat *CDKN2A*-deleted tumors, the use of PARP inhibitors to treat *STAG2*-mutant tumors, and the use of SYK inhibitors to treat *RB1*-mutant tumors (*Bailey et al., 2014*; *Gao et al., 2015b*; *Zhang et al., 2012*). Treatment with targeted agents significantly alters the cellular epigenetic and genetic landscape, often culminating in the development of resistance to the applied therapies (*Holohan et al., 2013*). We speculate that secondary alterations that tumors evolve to tolerate these drugs could also alter or blunt the aggressive phenotype caused by the original driver alteration. In this way, targeting a biomarker that confers poor prognosis could both directly lead to improved patient outcomes by triggering a robust clinical response, and indirectly help patients by forcing tumor evolution away from dependence on a driver of aggressive disease.

## Materials and methods

### Data sources

Patient cohorts analyzed in this study are listed in *Supplementary file 1*. For the TCGA analysis, preprocessed files from the Broad Institute TCGA Firehose were used (https://gdac.broadinstitute.org/). For the TCGA genomic copy number analysis, we used the HG19 segmented SCNAs, corrected for germline SCNAs. Overall survival time was used as a clinical endpoint for all cancer types except PRAD. Overall survival was chosen because it reflects an objective and unambiguous event, it is the gold-standard for oncology clinical trials, and it is widely-available across different studies (*Driscoll and Rixe, 2009*). However, as fewer than 2% of the patients in the PRAD cohort died during the follow-up period, 'days to biochemical recurrence' was used as a surrogate endpoint. For all cancers, survival or follow-up time from diagnosis were corrected for the days to sample procurement. Primary tumors (indicated with a '01' in the patient barcode) were used for every cancer type except SKCM; for this cancer, few primary samples were available, so metastatic samples (indicated with a '06' in the barcode) were included for patients in which no primary tumor was available. For additional discussion of the TCGA samples, see Supplemental Text 2. Pathology-assessed tumor cell fraction was obtained from the TCGA clinical files under 'Percent_tumor_cells'. Tumor stage and grade were similarly obtained from the appropriate TCGA clinical files.

Mutation, copy number, and clinical data from Release 25 of the International Genome Consortium were downloaded from the ICGC Data Portal (*Zhang et al., 2011*). Overall survival was used as a clinical endpoint for all cohorts except EOPC-DE; due to the few deaths in this cohort, recurrence-free survival was used as an endpoint. Cohorts were chosen based on the availability of WGS or WES data, and were included if they came from a cancer type comparable to the types that were studied in our TCGA analysis.

Copy number, mutation, and clinical data from cBioportal were downloaded as pre-processed files from www.cbioportal.org (*Gao et al., 2013*). For the patients described in *Zehir et al. (2017)* (the cBioportal/MSKCC_2017 cohorts), only primary tumors were included for all cancer types except melanoma.

## Overall analysis strategy

All processing and analysis was performed using Python. Cox proportional hazard analysis used the R survival package (https://cran.r-project.org/web/packages/survival/index.html) to compute Z scores and p values. Justification and further explanation for the use of Cox proportional hazards modeling can be found in Supplemental Text 1. The rpy2 project was used to control R from python, allowing seamless integration of Z score calculations with data processing and pan-cancer analysis. Pandas DataFrames were used as the primary structure for storing and manipulating data. Additionally, native numpy methods and arrays were for used occasionally for efficiently storing strictly numerical data, for example, as input to Cox proportional hazards models. The statsmodels package (www.statsmodels.org) was used for false discovery correction using the Benjamini-Hochberg procedure. Microsoft Excel was occasionally used for final data processing and examination, so a single apostrophe was added before gene names in intermediate data processing steps to protect genes from auto-formatting (*Zeeberg et al., 2004*).

Code was structured to allow ease of internal reuse and reproducibility of results. Cox univariate proportional hazards, Cox multivariate proportional hazards, Kaplan-Meier, and Stouffers analysis methods were factored into an analysis library, taking as input the data required to perform the computation as numpy arrays or pandas DataFrames.

## TCGA analysis

In addition to the code for statistical analyses, code for processing TCGA clinical files was factored into a common library. This approach allowed the same TCGA clinical file processing code to be executed across a variety of platform analyses, ensuring identical behavior for each platform. The TCGA clinical processing code selected the relevant clinical endpoints and sample procurement data. The processing translated the available clinical data into the required format for Cox proportional hazard models: an endpoint/survival time value and a censor value for each patient. Code to select tumor samples based on cancer type was also included in this library.

Raw input data for the mutation analysis needed additional preprocessing before Cox proportional hazard models could be constructed. This preprocessing included removing per-patient headers throughout the data and some data transposition. For all analyses using TCGA mutation data, mutations annotated as silent were excluded. Genes were only included in downstream analyses if they were mutated in 2% or more of the patients in a cancer type cohort.

Raw input data for copy number analysis also required substantial preprocessing. Copy number input data consists of per-patient, per-chromosome location maps of copy numbers (hg19 downloaded from the UCSC Genome Browser; *Tyner et al., 2017*). These maps were converted to a single copy number value for each gene. We created an interval tree (using the intervaltree python package, https://pypi.python.org/pypi/intervaltree) of the location maps for each chromosome and used the appropriate HGNC to convert chromosome locations to genes for each patient. We used the gene's transcriptional start site position to look up in the interval tree the copy number value for a gene. This analysis produced an intermediate file of a similar form to the other TGCA platforms, which allowed for straightforward Cox analysis. Note that Cox proportional hazards models are a threshold-independent method of performing survival analysis, and so no minimum or maximum threshold for a copy number alteration was specified.

A tumor was defined as having a focal amplification or deletion if its copy number was greater than 0.3 or less than −0.3, and the chromosomal interval with a copy number greater than 80% of the copy number at the gene of interest was less than or equal to 3 Mb (*Krijgsman et al., 2014*).

To calculate the number of structural alterations per tumor, the number of distinct copy number values per chromosome in the DNA segmentation file was summed for each patient.

## Pan-cancer TCGA analysis

For each platform and analysis type, we performed a pan-cancer analysis. This analysis created a single Z score for each gene by combining the per gene Z scores from each cancer type using Stouffer's method. To perform Stouffer's method, we took the sum of the Z scores for a single gene and divided that sum by the square root of the number of cancer types with Z scores for the gene (*Stouffer, 1949*). This meta-Z score was then compared against meta-Z scores obtained similarly from other platform analyses.

## Additional TCGA mutation analyses

We performed several additional analyses on mutation data, including double mutation combination Z scores, hotspot codon Z scores, and Z scores corrected for VAFs. For double mutation Z scores, we took the top 30 most common cancer driver genes and performed pairwise combinations. We then calculated Cox proportional hazards for each pair of genes, where a patient was considered to have a pairwise mutation if and only if both genes were non-silently mutated for that patient. Z scores were only calculated for a pair if (1) neither gene in the pair was statistically significant alone in the univariate analysis and (2) if both genes were mutated together in at least 10 patients.

Per-codon Z scores were calculated for a selected set of hotspot codons. Most cancer types were available in HG37, so HG37 mutation positions were used to locate codons. Mutations for OV and COADREAD were only available in HG36, so gene positions were converted to HG37 before codon processing. Per codon Z scores were calculated by first identifying patients with mutations in the relevant gene, then selecting from that set of patients those whose mutations were in the codon of interest. If 2% of patients or more had mutations in the selected codon, a Z score was calculated.

VAFs were calculated for 10 of the TCGA cancer types. We analyzed VAF data in two ways. First, we calculated Z scores, only counting a gene as mutated if its VAF was greater than or equal to 0.4. Secondly, we identified the median VAF score per gene, and calculated Z scores only counting a gene as mutated if its VAF was equal to or greater than the median VAF for that gene.

## CBioPortal analysis

CBioPortal was structured similarly to the TCGA analyses, though data processing was not factored into an independent library since each of these datasets was only used in one analysis. Copy number data from one CBioPortal cancer type, blca_mskcc, required initial preprocessing in the manner described above for TCGA copy numbers. Mutations were included if they were annotated as one of these types: In_Frame_Ins, Nonstop_Mutation, Translation_Start_Site, In_Frame_Del, Splice_Region, Frame_Shift_Ins, Frame_Shift_Del, Splice_Site, Nonsense_Mutation, or Missense_Mutation.

## ICGC analysis

ICGC analysis was structured similarly to CBioPortal analysis. Mutations were only included in downstream analyses if they were annotated as one of these types: disruptive inframe deletion, disruptive inframe insertion, frameshift variant, inframe deletion, missense variant, splice acceptor variant, splice donor variant, stop gained, or stop lost. Z scores were calculated if a gene was mutated in 2% or more of the patients in a particular cohort.

## Identification of high-confidence biomarkers associated with drug sensitivities

Across independent datasets, cohorts of patients from related cancer types were identified. Mutations or CNAs significantly associated with patient prognosis ($Z > 1.96$ or $Z < -1.96$) in two or more independent cohorts from each cancer type were determined. Then, the subset of these alterations that remained highly-significant ($Z > 3.3$ or $Z < -3.3$) across all cohorts from the same cancer type were classified as high-confidence biomarkers. In some instances, amplifications that spanned continuous chromosomal regions were found to correlate with patient prognosis. These segments were identified manually. For the determinations of therapeutic sensitivity described below, the gene with the minimum meta-Z score (for deletions) or maximum meta-Z score (for amplifications) within a segment was chosen to represent the segment as a whole.

Therapeutic sensitivity data for PDXs was acquired from (*Gao et al., 2015a*). To identify mutations that correlated with therapy sensitivity, for each drug or drug combination, a comparison was performed if five or more PDXs had a mutation in a gene of interest, and if five or more PDXs were wild-type for a gene of interest. For genes and therapies fitting these criteria, we next identified instances in which the therapy resulted in a clinical response in the mutant population, defined as an average 'Best Average Response'<15% tumor growth among PDXs with a mutation in the gene of interest. Finally, for genes and therapies fitting these criteria, we performed a t-test for the 'Best Average Response' between PDXs with mutant and wild-type copies of a gene of interest. We reported therapies in which these criteria were met and tumors with mutation were more sensitive to the therapy than tumors with wild-type copies of the gene of interest ($p < 0.01$).

To identify CNAs that correlated with therapy sensitivity in the PDX cohort, amplifications and deletions (CNA >|.3|) were called, and then considered separately. As above, CNAs were included if five or more PDXs exhibited an alteration, and if five or more PDXs did not exhibit that alteration. For genes and therapies fitting these criteria, we next identified instances in which the therapy resulted in a clinical response in the altered population, defined as an average 'Best Average Response'<15% tumor growth among PDXs with an amplification or deletion in the gene of interest. Finally, for genes and therapies fitting these criteria, we performed a t-test for the 'Best Average Response' between PDXs with mutant and wild-type copies of a gene of interest. We reported therapies in which these criteria were met and tumors with a mutation were more sensitive to the therapy than tumors with wild-type copies of the gene of interest (p < 0.01).

Therapeutic sensitivity data from cancer cell lines was acquired from (*Iorio et al., 2016*). For this data, two different comparisons were used. First, the calculations described below were performed for cell lines from the specific cancer type that the high-confidence biomarker was identified in. If this analysis yielded no significant vulnerabilities, then the calculations were repeated across all cancer types (pan-cancer).

High-confidence mutations were assessed if five or more cell lines in the set of interest had a nonsynonymous mutation in that gene, and if five or more cell lines had wild-type copies of that gene. CNAs were assessed if five or more cell lines had an alteration (deletion or amplification) of that gene, and if five or more cell lines lacked that alteration. For each comparison, T-tests were performed between the log(IC50) value of every tested compound. For single-cancer type analyses, a threshold of p < 0.01 was used to identify significance, while for pan-cancer analyses, a threshold of p < 0.0001 was used to identify significance.

## Code

Code is available on GitHub at https://github.com/joan-smith/genomic-features-survival (*Smith, 2018*; copy archived at https://github.com/elifesciences-publications/genomic-features-survival).

## Kaplan-Meier analysis

Kaplan-Meier plots were generated using Graphpad Prism. Deletions and amplifications in Kaplan-Meier plots correspond to CNAs > |0.3|; deep deletions and high-copy gains correspond to CNAs > |1|. P values reported in KM plots were generated by the log-rank test in Prism. Note that Kaplan-Meier plots are displayed in this manuscript primarily for the ease of visualizing patient outcomes. Z scores were always generated with Cox proportional hazards modeling, which does not require the selection of artificial cut-offs or thresholds for continuous data.

## Additional data sources and tools

The 30 frequently-mutated cancer driver genes were acquired from (*Zehir et al., 2017*). NCI-SEER statistics were downloaded from https://seer.cancer.gov. Total tumor aneuploidy scores, ABSO-LUTE-determined purity values, and leukocyte infiltration was obtained from (*Taylor et al., 2018*). Hyper-mutated samples were obtained from (*Bailey et al., 2018*). Lollipop plots were generated using Lollipops software (*Jay and Brouwer, 2016*). Density plots were generated with Python scripts using matplotlib (https://matplotlib.org/). Single base-pair mutations were mapped to codons using PolyPhen-2 (*Adzhubei et al., 2010*).

## Supplemental text 1. Cox proportional hazards modeling

Multiple statistical techniques have been developed to perform survival or 'time-to-failure' analysis (reviewed in *Kleinbaum and Klein, 2012*). These include Kaplan-Meier analysis, Cox proportional hazards regression, accelerated failure time modeling, and many others. In this paper, we chose to apply Cox proportional hazards regression to analyze cancer survival data. The Cox model is represented by the following function:

$$h(t,X) = h_0(t)e^{\sum_{i=1}^{n} \beta_i X_i}$$

Where t is the survival time, h(t, X) is the hazard function, $h_0(t)$ is the baseline hazard, $X_i$ is a potential prognostic variable, and $\beta_i$ indicates the strength of the association between a prognostic

variable and survival. In this model, patients have a baseline, time-dependent risk of death [$h_0(t)$], modified by time-independent prognostic features that either increase ($\beta_i > 0$) or decrease ($\beta_i < 0$) risk of death. In this paper, we report Z scores, which are calculated by dividing the regression coefficient ($\beta_i$) by its standard error.

Cox proportional hazards modeling was chosen for several reasons. First, unlike Kaplan-Meier analysis, Cox models do not require the selection of a threshold or cut-off, so continuous data like gene expression values do not need to be dichotomized. (Note that in this manuscript, Kaplan-Meier plots are provided for visualization purposes, but the reported Z scores are always from Cox models). Secondly, Cox models can accept both continuous and discrete input data, allowing this approach to be used to analyze both binary (e.g., mutant vs. non-mutant) and continuous (e.g., gene copy number) genomic features. Thirdly, Cox models are amenable to both univariate (i = 1) and multivariate (i > 1) analyses. Fourthly, Cox regression allows us to calculate Z scores and a p value for each association, as Z scores represent the number of standard deviations from the mean of a normal distribution. Fifthly, Z scores encode the directionality of an association: poor prognostic factors will exhibit $\beta_i$ values greater than 0, while favorable prognostic factors will exhibit $\beta_i$ values less than 0. This allows 'positive' and 'negative' survival features to be directly compared. Sixthly, Z scores are useful for meta-analyses, as they can be combined using Stouffer's Method (*Stouffer, 1949*):

$$Z = \frac{\sum_{i=1}^{n} Z_i}{\sqrt{k}}$$

Seventhly, Cox proportional hazards modeling is commonly used in both previous genome-wide survival analyses and in numerous clinical biomarkers studies (*Dhanasekaran et al., 2001*; *Fukuoka et al., 2011*; *Gentles et al., 2015*; *Parker et al., 2009*; *Wang et al., 2005*), facilitating comparison with other biomarker discovery efforts.

To verify the underlying normality of the Z scores, we generated qq plots for gene copy number values (*Figure 1—figure supplement 2C*). The resulting distributions for CNAs were generally linear, as expected, with occasional shoulders at low and high Z scores. We similarly calculated Z scores for all genes harboring coding-sequence mutations; however, we discovered that this resulted in plateaus around the origin in multiple cancer types. These aberrations were caused by the occurrence of rare, random mutations in multiple genes that lacked any prognostic power. To eliminate these plateaus, we experimented with different thresholds for mutational analysis. Considering only mutations that occurred in a certain percentage of cancer patients diminished the appearance of the plateaus, but high thresholds also eliminated from consideration mutations in a number of known cancer drivers. We selected a 2% threshold to balance between maintaining the normality of the Z score distribution while also retaining infrequent but significant mutations in driver genes.

Note that in many survival analysis papers, a 'feature selection' step is included to identify a minimal number of features that can accurately identify at-risk patients. We performed an unbiased, whole-genome analysis without feature selection, to generate a Z score for every gene and for every feature type in the genome. No feature selection step is applied in this work.

## Supplemental text 2. Survival analysis in TCGA cohorts

Patient cohorts that were assembled for the TCGA were collected in order to allow a molecular analysis of the major cancer subtypes found within the United States. Though clinical information was collected for nearly all patients, these cohorts were not specifically chosen in order to conduct survival studies. We posit that our survival analysis is appropriate for several reasons. First, we verified that the overall survival times of patients within the TCGA is highly consistent with national epidemiological data collected by the NCI (*Figure 1—figure supplement 2D–E*). Secondly, we found that many well-established biomarkers hold prognostic significance in TCGA cohorts, including *IDH1* mutations in glioma (*Figure 1—figure supplement 7*), *TP53* mutations in breast cancer (*Figure 1—figure supplement 3*), tumor stage and grade in multiple cancer types (*Figure 2—figure supplement 3*), and more. Thirdly, we validated the survival patterns that we describe in the TCGA in several independent patient cohorts, indicating that these are not TCGA-specific phenomena (*Figure 4*). Fourthly, in an independent analysis of the quality of clinical annotations in the TCGA (*Liu et al., 2018*), none of the cohort/endpoint combinations chosen for this study were classified as

'not recommended for use.' Fifthly, our efforts build upon a robust body of work that has also performed survival analyses on TCGA cohorts, and, in some cases, similarly validated findings from the TCGA in independent patient populations (*Andor et al., 2016*; *Davoli et al., 2017*; *Gentles et al., 2015*; *Guinney et al., 2015*; *Uhlen et al., 2017*). Finally, we note that the TCGA has several benefits over standard investigator-initiated survival studies. Patient samples were collected and analyzed in an unbiased manner, precluding the possibility of the 'file-drawer problem' (failing to publish negative results) or post-hoc sample size adjustment (ending patient enrollment when a significant result is found). Significantly more molecular data is available from TCGA tumors than in any other comparably-sized dataset, which allows for multivariate and correlational analyses of different facets of tumor genomes. All data from the TCGA and all code from this manuscript are publicly-available, allowing easy replication and extension upon this analysis.

## Acknowledgments

We thank members of the Sheltzer Lab for helpful comments on this work. Research in the Sheltzer Lab is supported by an NIH Early Independence Award (1DP5OD021385), a Breast Cancer Alliance Young Investigator Award, and a CSHL-Northwell Translational Cancer Research Grant.

## Additional information

### Competing interests

Joan C Smith: affiliated with Google Inc. The author has no financial interests to declare. The other author declares that no competing interests exist.

### Funding

| Funder | Grant reference number | Author |
|---|---|---|
| National Institutes of Health | 1DP5OD021385 | Jason M Sheltzer |
| Breast Cancer Alliance | Young Investigator Award | Jason M Sheltzer |
| Cold Spring Harbor Laboratory | CSHL-Northwell Translational Cancer Research Grant | Jason M Sheltzer |

The funders had no role in study design, data collection and interpretation, or the decision to submit the work for publication.

### Author contributions

Joan C Smith, Conceptualization, Software, Formal analysis, Investigation, Methodology, Writing—review and editing; Jason M Sheltzer, Conceptualization, Resources, Formal analysis, Supervision, Funding acquisition, Investigation, Visualization, Methodology, Writing

### Author ORCIDs

Jason M Sheltzer (iD) https://orcid.org/0000-0003-1381-1323

### Decision letter and Author response

Decision letter https://doi.org/10.7554/eLife.39217.038
Author response https://doi.org/10.7554/eLife.39217.039

## Additional files

### Supplementary files

• Supplementary file 1. Cancer survival cohorts analyzed in this study.
DOI: https://doi.org/10.7554/eLife.39217.024
• Supplementary file 2. Cox proportional hazards modeling of mutations in the TCGA.

DOI: https://doi.org/10.7554/eLife.39217.025

• Supplementary file 3. Cox proportional hazards modeling of CNAs in the TCGA.
DOI: https://doi.org/10.7554/eLife.39217.026

• Supplementary file 4. Cox proportional hazards modeling in the TCGA adjusted for stage, grade, or subtype.
DOI: https://doi.org/10.7554/eLife.39217.027

• Supplementary file 5. Cox proportional hazards modeling adjusted for TP53 status or total aneuploidy.
DOI: https://doi.org/10.7554/eLife.39217.028

• Supplementary file 6. Cox proportional hazards modeling of cancer cohorts from the ICGC or curated by cBioportal.
DOI: https://doi.org/10.7554/eLife.39217.029

• Supplementary file 7. Cox proportional hazards modeling of the MSKCC_2017 cohorts.
DOI: https://doi.org/10.7554/eLife.39217.030

• Supplementary file 8. High-confidence biomarkers and their associated therapeutic sensitivities.
DOI: https://doi.org/10.7554/eLife.39217.031

• Transparent reporting form
DOI: https://doi.org/10.7554/eLife.39217.032

## Data availability

Data is publicly available from The Cancer Genome Atlas, the International Cancer Genome Consortium, and the other resources listed in the Materials and Methods section. Code is available on GitHub at https://github.com/joan-smith/genomic-features-survival (copy archived at https://github.com/elifesciences-publications/genomic-features-survival).

The following previously published datasets were used:

| Author(s) | Year | Dataset title | Dataset URL | Database and Identifier |
|---|---|---|---|---|
| Broad Institute TCGA Genome Data Analysis Center | 2016 | Broad Institute TCGA Firehose stddata__2016_01_28 | http://gdac.broadinstitute.org/runs/stddata__2016_01_28/ | Broad GDAC Firehose, 10.7908/C11G0KM9 |
| International Cancer Genome Consortium | 2017 | International Cancer Genome Consortium | https://dcc.icgc.org/releases/release_25 | ICGC Data Portal, Release 25 |

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
