## [Decision Letter]

Thank you for submitting your article "Genetic determinants of cancer patient outcome" for consideration by *eLife*. Your article has been reviewed by Dr. Jeffrey Settleman as the Senior Editor, a Reviewing Editor, and three reviewers.

The reviewers have discussed the reviews with one another, and I have drafted this decision to help you prepare a revised submission. The reviewers feel that your paper should be accepted for publication after our concerns are addressed, either by performing additional analyses or by qualifying your findings in the writing.

Summary:

The manuscript by Sheltzer and coworkers "Genetic determinants of cancer patient outcome" provides an extensive evaluation of the prognostic information associated with frequent mutations and copy number alterations across multiple tumors from the TCGA and other validation sets. While overall the findings are intriguing, the reviewers have suggestions to correct gaps in this evaluation. If the authors can address the issues noted below however this could be a very interesting article.

We have the following suggestions to improve the manuscript.

1) The observation that copy number (CN) of driver genes and not mutations of the same genes associate with outcome is interesting and intriguing. However, the authors should clearly point out that their current analysis addresses whether CN of driver genes or mutations of various genes have inherent prognostic value in patients treated with standard chemotherapy regimens.

The authors need to state explicitly that their analysis does not address whether newer therapies targeted to inhibit specific activated oncoproteins (e.g. mutant EGFR in lung cancer) will ultimately have an important impact. This analysis must await adequate follow-up of patients treated with targeted drugs. The same caveat applies to oncogenes included in CN such as MDM2. Specific inhibitors may be especially active in these patients, but this will have to be addressed in future studies.

2) The authors should make some attempt to consider the impact of disease-specific mutations. For example, mutations occurring in >2% of patients are highlighted. However, EGFR mutations are more common in lung cancer than they are in other solid tumor types. I also note that specific variants known to influence function are not accounted for – noting EGFR as an example, there is no acknowledgment of exon 19 del or L858R mutations (as opposed to variants of unknown clinical significance). If these types of analyses are beyond the scope of what can be done in two months, the authors should clearly state that their paper is meant to highlight the role of CN in prognosis and acknowledge that disease- and mutation-specific analyses will be needed to assess the importance of mutations that activate specific oncoproteins in specific types of cancer.

3) Different statistical power for CN and mutations: perhaps the overall frequencies of mutations are too low and underpowered as compared to CN event frequencies (probably not the case for TP53 but for most other genes), thus explaining the lower number of statistically significant associations. In addition, a more meaningful comparison could be the hazards ratio from the Cox model, which would be independent of the number of samples with CN/mutations. Reporting z-scores (analogously to p-values) for the Cox regression is necessary to reject the null hypothesis but it would be informative to also provide hazards ratio and confidence interval for the Cox hazards regression (These data can be easily obtained from the R survival package Cox implementation even for continuous variables such as CN).

4) Gain/loss of function mutations and CN: From the data presented (Figure 3D) it is unclear whether gene focal CN and mutations overlap or tend to be mutually exclusive. A more exhaustive exploration of gain/loss of function through the directionality of prognostic associations would be desirable: It is interesting that TP53 shows significant and inverse association with survival in the METABRIC set (Figure 4J); that might be an expected behavior for tumor suppressor genes with loss of function mutations; As opposed to gain of function oncogene hot spots (BRAF600, RAS12, etc.), which should show positive, z-score for both CN and mutation. Is this directionality affecting the results presented in Figure 3D were EGFR shows loss of significance for the combined CN+ mutation bar? An oncoprint/table showing frequencies mutations overlaying CN would also be helpful to understand gain/loss of function events. Along these same lines, how do the authors know for sure which genes are the drivers within individual CN regions. Couldn't the affect on prognosis of some CN regions be based on the simultaneous loss or gain of function of two or more linked genes within a given region?

5) The approach to quantify aneuploidy (Taylor et al., 2018) didn't report any association with outcome in their original manuscript, despite previous literature reports (Kallioniemi et al., 1987; Kokal et al., 1986; Friedlander et al., 1984; Merkel and McGuire, 1990; Zimmerman et al., 1987) and only weak associations were found in this manuscript (Figure S10B). This measure of aneuploidy quantifies whole chromosome and whole arm numerical changes but doesn't account for focal events or other elements (i.e. structural variants) associated with chromosomal instability. As a consequence, the question of whether CN of driver genes carry prognostic information independently of overall chromosomal instability burden remains unanswered. The authors should be able to quantify the number of focal events per sample as an alternative measure of chromosomal instability and evaluate whether the prognostic capabilities of driver genes CN remain significant. Another measure of chromosomal instability could be the number of CN breakpoints (per Mbase) that can be obtained from segmentation data from TCGA samples.

6) Patient risk stratification is a highly complex, specialized and histology-type specific area of research. It would not be within the scope of this manuscript to review each tumor type strategy, however it would be useful if the authors could identify one or two real case scenarios in which their findings could potentially be integrated into clinical diagnostic practice (i.e., how MDM4 gain of copy number could be used to complement the Gleason score in predicting risk in prostate cancer).

The authors are also using outcomes data without providing definitions of how they were collected, though they do provide very detailed rationale for the choice of methods (Cox modeling in particular).

7) Copy number analysis is interesting but would benefit from a more in-depth look by disease type and perhaps by known oncogenic drivers within those cancer types. Pooling these data here may be missing critical signals within subsets. The need for further analysis by cancer type and histology should at least be acknowledged in the Discussion section if it cannot be completed within the timescale. Prognostic information based on gene or copy number depends to some extent on treatment. The authors should address the treatment that patients in this analysis received to the extent that they are able. Were any of the patients treated with modern day TKIs and immunotherapy?

[Editors' note: further revisions were requested prior to acceptance, as described below.]

Thank you for resubmitting your work entitled "Genetic determinants of cancer patient outcome" for further consideration at *eLife*. Your revised article has been favorably evaluated by Jeffrey Settleman (Senior Editor), a Reviewing Editor, and two reviewers.

The manuscript has been improved but there are some remaining issues that need to be addressed before acceptance, as outlined below:

Can you please revise the Title to reflect more specifically the key conclusion regarding gene copy number information. Otherwise, it sounds like the title of a review article.

*Reviewer #1:*

The paper now meets my expectations and should be accepted as is.

*Reviewer #3:*

The authors thoroughly addressed the comments raised by reviewers and therefore my evaluation is positive and recommend for publication.

---

## [Author Response]

We have the following suggestions to improve the manuscript.1) The observation that copy number (CN) of driver genes and not mutations of the same genes associate with outcome is interesting and intriguing. However, the authors should clearly point out that their current analysis addresses whether CN of driver genes or mutations of various genes have inherent prognostic value in patients treated with standard chemotherapy regimens.The authors need to state explicitly that their analysis does not address whether newer therapies targeted to inhibit specific activated oncoproteins (e.g. mutant EGFR in lung cancer) will ultimately have an important impact. This analysis must await adequate follow-up of patients treated with targeted drugs. The same caveat applies to oncogenes included in CN such as MDM2. Specific inhibitors may be especially active in these patients, but this will have to be addressed in future studies.

We agree with the reviewers that certain mutations may harbor prognostic information through their ability to predict sensitivity to targeted agents. In the Discussion section, we have added the following: “Secondly, most patients in the TCGA cohorts were treated with standard cytotoxic drugs. As targeted and immuno-therapies are increasingly adopted in the clinic, oncogenic mutations that were non-prognostic in the datasets analyzed here may be able to predict sensitivity to specific therapeutic agents.”

2) The authors should make some attempt to consider the impact of disease-specific mutations. For example, mutations occurring in >2% of patients are highlighted. However, EGFR mutations are more common in lung cancer than they are in other solid tumor types. I also note that specific variants known to influence function are not accounted for – noting EGFR as an example, there is no acknowledgment of exon 19 del or L858R mutations (as opposed to variants of unknown clinical significance). If these types of analyses are beyond the scope of what can be done in two months, the authors should clearly state that their paper is meant to highlight the role of CN in prognosis and acknowledge that disease- and mutation-specific analyses will be needed to assess the importance of mutations that activate specific oncoproteins in specific types of cancer.

In general, mutations in cancer driver genes in the TCGA cohorts are very likely to affect gene function. For instance, across all 16 cancer types, 70% of mutations in KRAS are in a single codon (c12), while another 10% of mutations are in codon 13 – both clearly KRAS-activating. Thus, by considering all KRAS mutations together, we do not believe that we are diluting a signal in the data by conflating KRAS-activating mutations with mutations that fail to affect protein function. Nonetheless, we have made several attempts to identify codon-specific and tissue-specific mutations with prognostic power:

1) We identified the 25 most-frequently mutated codons in the TCGA cohorts (KRAS^c12^, BRAF^c600^, etc.) and performed Cox modeling to test whether they were prognostic. Out of 480 possible mutation/cancer type combinations, only 6 achieved statistical significance (Figure 1—figure supplement 4D). Moreover, the most significant single-codon mutation (IDH1^c132^ in glioblastoma) recapitulated what was observed when all IDH1 mutations were considered together (Figure 1—figure supplement 7A).

2) We identified all “hotspot” mutations in cancer: specific codons that were mutated in five or more patients. We pooled patients who harbored a “hotspot” mutation in a specific gene, while eliminating from consideration mutations that failed to reach this threshold. Thus, for KRAS, we pooled patients with mutations in codon 12, 13, 61, etc., but eliminated the single patients with mutations in codon 8 and codon 22. However, when we performed Cox modeling considering only patients who had mutations in a “hotspot” codon, we failed to detect any significant prognostic markers outside of GBMLGG (Figure 1—figure supplement 4E).

3) We identified the most common “hotspot” mutations in each specific cancer type: codons that were mutated in at least 4% of patients within an individual cohort. This identified recurrent mutations that were observed across cancer types (like PIK3CA^C545^) and mutations that were unique to specific cancer types (like FGFR3^c249^ mutations in BLCA and GNAS^c844^ mutations in PAAD). Univariate Cox modeling again revealed very few prognostic alterations among these specific mutations (3 out of 66, when GBMLGG is excluded; Figure 1—figure supplement 4F). The EGFR^L858^ alteration that the reviewers inquired about fell below our cut-off, but analyzing it separately we found that this alteration was also non-prognostic in this cohort.

Thus, while we agree that larger cohorts may yield additional prognostic mutations, using several different analytical approaches, we have been unable to identify robust codon-specific or disease-specific prognostic mutations, outside of glioma/glioblastoma.

3) Different statistical power for CN and mutations: perhaps the overall frequencies of mutations are too low and underpowered as compared to CN event frequencies (probably not the case for TP53 but for most other genes), thus explaining the lower number of statistically significant associations. In addition, a more meaningful comparison could be the hazards ratio from the Cox model, which would be independent of the number of samples with CN/mutations. Reporting z-scores (analogously to p-values) for the Cox regression is necessary to reject the null hypothesis but it would be informative to also provide hazards ratio and confidence interval for the Cox hazards regression (These data can be easily obtained from the R survival package Cox implementation even for continuous variables such as CN).

It is conceivable that different statistical power between copy number and mutations could contribute to these results. For instance, as mutations are discrete (“mutant” vs. “wild-type), while CNA values are continuous, every copy number measurement contains inherently more information than a mutation in our Cox models. To test whether this difference in statistical power contributed to our conclusions, we discretized the copy number measurements, dividing them into “deletions” (<-0.3), “copy-neutral” (≥-0.3 and ≥0.3), and “amplifications” (>0.3). Though this procedure significantly reduced the information contained in every CNA, it did not greatly affect our overall conclusion. Among the 30 most frequently mutated genes in the 16 TCGA cohorts, we detected 23 prognostic mutations (Figure 1C) compared to 108 prognostic CNAs (Figure 2B). After we discretized the copy number data, 94 CNAs were still significant. This result is now included in Figure 2—figure supplement 2A.

As per the reviewers’ request, we have now included hazard ratio calculations in several areas in this manuscript (Figure 1—figure supplement 3, Figure 5—figure supplement 1, Supplementary file 2B, Supplementary file 3B). However, relying on hazard ratios is tricky due to the difficulty in directly comparing them or extracting useful information from them. For instance, in multivariate models from the TCGA breast cancer cohort, age is significantly associated with poor outcome (Z = 2.97, P <.003), but the hazard ratio for this risk factor is only 1.03 (Figure 5—figure supplement 1A). Additionally, we calculated hazard ratios for our main 30 gene x 16 cohort analysis – but for 22 of these calculations, the hazard ratio was 0, with a 95% CI from 0 to infinity, as no patients with the indicated mutation died during data collection. Thus, in this paper, we have relied on Z scores in order to test the null hypothesis that a given variable is uninformative in cancer prognosis.

4) Gain/loss of function mutations and CN: From the data presented (Figure 3D) it is unclear whether gene focal CN and mutations overlap or tend to be mutually exclusive. A more exhaustive exploration of gain/loss of function through the directionality of prognostic associations would be desirable: It is interesting that TP53 shows significant and inverse association with survival in the METABRIC set (Figure 4J); that might be an expected behavior for tumor suppressor genes with loss of function mutations; As opposed to gain of function oncogene hot spots (BRAF600, RAS12, etc.), which should show positive, z-score for both CN and mutation. Is this directionality affecting the results presented in Figure 3D were EGFR shows loss of significance for the combined CN+ mutation bar? An oncoprint/table showing frequencies mutations overlaying CN would also be helpful to understand gain/loss of function events. Along these same lines, how do the authors know for sure which genes are the drivers within individual CN regions. Couldn't the affect on prognosis of some CN regions be based on the simultaneous loss or gain of function of two or more linked genes within a given region?

The relationship between the directionality of a Z score and the function of a particular gene is complicated. For cancer mutations, it does not appear that there is a strong link between the two. For instance, IDH1 mutations are clearly oncogenic, but they are associated with a highly significant negative Z score (as it has been recognized that IDH1-mutant gliomas are less aggressive than gliomas driven by other alterations; Yan et al., 2010). We’ve also found that tumor suppressor mutations can be associated with both positive Z scores (e.g., RB1 in BLCA) and negative Z scores (e.g., *FBXW7* in COADREAD; Figure 2B).

For CNAs, we definitely note the presence of tumor suppressors in Z-score valleys (*CDKN2A* on 9p, *PTEN* on 10q) and oncogenes in Z-score peaks (*PIK3CA* on 3q, MYC on 8q; Figure 2C). But we fully agree with the reviewers that multiple genes within an amplified or deleted region are likely to affect the functional consequences of any given CNA. We are hesitant to draw strong conclusions about specific drivers from fundamentally correlative observations, other than the few well-characterized examples that we have highlighted. In our lab, we are working to develop CRISPR-based tools to manipulate the copy number of DNA segments, which we hope to use to dissect these recurrently-altered regions.

Concerning the overlap between mutations and CNAs: in some instances, there’s significant overlap, in other instances there’s not. For instance, in PRAD, 71% of tumors that have a PTEN mutation also have a PTEN deletion, likely reflecting a segmental LOH event. In contrast, among COADREAD tumors with ARID1A mutations, only 8% have a deletion in ARID1A, suggesting that mutations and deletions are commonly mutually-exclusive methods of eliminating this gene. We haven’t been able to detect an overarching explanation as to when there is or is not overlap. Nonetheless, these instances of overlap do not affect the overall conclusion that 95% of prognostic CNAs remain associated with outcome in multivariate models that include gene mutation status, indicating that even changes in wild-type loci can affect cancer aggressiveness (Figure 3D).

5) The approach to quantify aneuploidy (Taylor et al., 2018) didn't report any association with outcome in their original manuscript, despite previous literature reports (Kallioniemi et al., 1987; Kokal et al., 1986; Friedlander et al., 1984; Merkel and McGuirer, 1990; Zimmerman et al., 1987) and only weak associations were found in this manuscript (Figure S10B). This measure of aneuploidy quantifies whole chromosome and whole arm numerical changes but doesn't account for focal events or other elements (i.e. structural variants) associated with chromosomal instability. As a consequence, the question of whether CN of driver genes carry prognostic information independently of overall chromosomal instability burden remains unanswered. The authors should be able to quantify the number of focal events per sample as an alternative measure of chromosomal instability and evaluate whether the prognostic capabilities of driver genes CN remain significant. Another measure of chromosomal instability could be the number of CN breakpoints (per Mbase) that can be obtained from segmentation data from TCGA samples.

We have now quantified the number of CN breakpoints from the TCGA segmentation data as a proxy for structural instability in each tumor. P53-mutant tumors harbor more breakpoints, as expected (Figure 3—figure supplement 1B), and the total breakpoint burden is a prognostic factor in 8/16 cohorts (Figure 3—figure supplement 1C). Nonetheless, 75% of driver gene alterations remain significantly-associated with outcome in multivariate models that include tumor breakpoint burden, providing further evidence that the prognostic power of these alterations is not an indirect consequence of their correlation with chromosomal instability (Figure 3—figure supplement 2E-F).

6) Patient risk stratification is a highly complex, specialized and histology-type specific area of research. It would not be within the scope of this manuscript to review each tumor type strategy, however it would be useful if the authors could identify one or two real case scenarios in which their findings could potentially be integrated into clinical diagnostic practice (i.e., how MDM4 gain of copy number could be used to complement the Gleason score in predicting risk in prostate cancer).The authors are also using outcomes data without providing definitions of how they were collected, though they do provide very detailed rationale for the choice of methods (Cox modeling in particular).

We believe that the high-confidence biomarkers we identified in the combined meta-analyses have the potential to be integrated into clinical practice. As suggested by the reviewers, we built multivariate models combining the top-scoring copy number alterations with standard prognostic criteria (receptor status in breast cancer, Clark score in melanoma, Gleason score in prostate cancer, etc.). The CNAs remained associated with survival time in these combined models, highlighting their potential clinical relevance (Figure 5—figure supplement 1).

Additionally, we have provided more information concerning our choice of clinical endpoints. In the Materials and methods section, we write: “Overall survival time was used as a clinical endpoint for all cancer types except PRAD. Overall survival was chosen because it reflects an objective and unambiguous event, it is the gold-standard for oncology clinical trials, and it is widely-available across different studies^84^. However, as fewer than 2% of the patients in the PRAD cohort died during the follow-up period, ‘days to biochemical recurrence’ was used as a surrogate endpoint.”

7) Copy number analysis is interesting but would benefit from a more in-depth look by disease type and perhaps by known oncogenic drivers within those cancer types. Pooling these data here may be missing critical signals within subsets. The need for further analysis by cancer type and histology should at least be acknowledged in the Discussion section if it cannot be completed in two months. Prognostic information based on gene or copy number depends to some extent on treatment. The authors should address the treatment that patients in this analysis received to the extent that they are able. Were any of the patients treated with modern day TKIs and immunotherapy?

We agree with the reviewers that further analysis within specific tumor histologies is warranted. In the Discussion section, we write, “We consider it possible that, with larger sample sizes or more-specific tumor subtypes, additional prognostic mutations could be identified.”

However, we think that it’s noteworthy that in the exact same cohorts in which driver gene mutations are generally non-prognostic, CNAs in those same genes are highly prognostic. And, as we continue to collect larger cohorts from discrete cancer subtypes, we continue to see CNAs out-perform mutations. For instance, consider the >2000-patient METABRIC breast cancer cohort that we used as a validation dataset, in which 2/25 common mutations are prognostic while 12/25 CNAs are prognostic (Figure 4J).

We do not yet know the degree to which the stratification power of CNAs will depend on treatment. Certain pathological findings – like positive surgical margins, lymph-node dissemination, and tumor pleomorphism – are broadly prognostic across a range of cancers, independent of subsequent treatment. It is possible that some CNAs could be similarly prognostic, regardless of the specific therapeutic regimen.

A small fraction of patients were treated with targeted therapies or immunotherapies. For instance, about 3% of melanoma patients are recorded as having received a BRAF inhibitor, while about 5% received a CTLA-4 antibody. In general, eliminating these patients from consideration does not substantially affect our results (Figure 1—figure supplement 5A and data not shown).

[Editors' note: further revisions were requested prior to acceptance, as described below.]

The manuscript has been improved but there are some remaining issues that need to be addressed before acceptance, as outlined below:Can you please revise the Title to reflect more specifically the key conclusion regarding gene copy number information. Otherwise, it sounds like the title of a review article.

We have changed our Title to “Systematic identification of mutations and copy number alterations associated with cancer patient prognosis.